# Guiding Skill Discovery with Foundation Models

## Abstract

Learning diverse skills without manually designed reward functions can greatly reduce human effort, making the process more autonomous and realistic. However, existing skill discovery methods focus solely on maximizing the diversity of skills without considering human preferences, which leads to undesirable behaviors and possibly dangerous skills. For instance, a cheetah robot trained using previous methods learns to roll in all directions to maximize skill diversity, whereas we would prefer it to run without flipping or entering hazardous areas. In this work, we propose a **Fo**undation model **G**uided (FoG) skill discovery method, which incorporates human intentions into skill discovery through foundation models. Specifically, FoG extracts a score function from foundation models to evaluate states based on human intentions, assigning higher values to desirable states and lower to undesirable ones. These scores are then used to re-weight the rewards of skill discovery algorithms. By optimizing the re-weighted skill discovery rewards, FoG successfully learns to eliminate undesirable behaviors, such as flipping or rolling, and to avoid hazardous areas in both state-based and pixel-based tasks. Interestingly, we show that FoG can discover skills involving behaviors that are difficult to define. Interactive visualisations are available from `https://sites.google.com/view/submission-fog`.

## 1 Introduction

Reinforcement learning (RL) has shown promising results in robotics (Tang et al., 2024; Wu et al., 2023) and games (Vasco et al., 2024; Zhang et al., 2024). Typically, RL requires carefully designed reward functions, which demand significant expert efforts (Schenck & Fox, 2018; Sowerby et al., 2022). In contrast, Unsupervised RL (Laskin et al., 2021; Rajeswar et al., 2023) aims to eliminate task-specific reward functions and train agents in a self-supervised manner. One key direction in unsupervised RL is pre-training agents to acquire diverse skills that can potentially be useful in downstream tasks (Eysenbach et al., 2018; Park et al., 2023b), termed unsupervised skill discovery. Most prior methods in unsupervised skill discovery focus on maximizing skill diversity, encouraging agents to achieve diversity in both low-level behaviors and high-level policies. For instance, a cheetah robot trained using previous methods (Park et al., 2022; 2023b) learns to flip or roll (low-level behavior) in all directions (high-level policy). However, wide motions like flipping or rolling could damage the robot, and entering restricted areas might pose safety risks. Ideally, we want agents to learn skills that are not only diverse, but also aligned with specific intentions, such as eliminating undesirable behaviors or avoiding certain areas.

To integrate human intentions into skill discovery, we introduce a **Fo**undation model **G**uided (FoG) method. More specifically, FoG (see Figure 1) utilizes foundation models (Radford et al., 2021; Ouyang et al., 2022; Bordes et al., 2024) to assign higher scores for desirable behaviors and lower for undesirable ones. These scores are then used to re-weight the rewards of unsupervised skill discovery algorithms. By optimizing these re-weighted rewards, FoG learns diverse skills while aligning with given human intentions. FoG stands out from previous methods by being more autonomous, as it does not rely on costly expert demonstrations like Kim et al. (2024b), and more versatile, as it works with both visual inputs and compact state information, unlike Rho et al. (2024), which requires precise ground-truth states.

Our main contributions are fourfold, and the FoG codebase can be found in the supplemental materials:

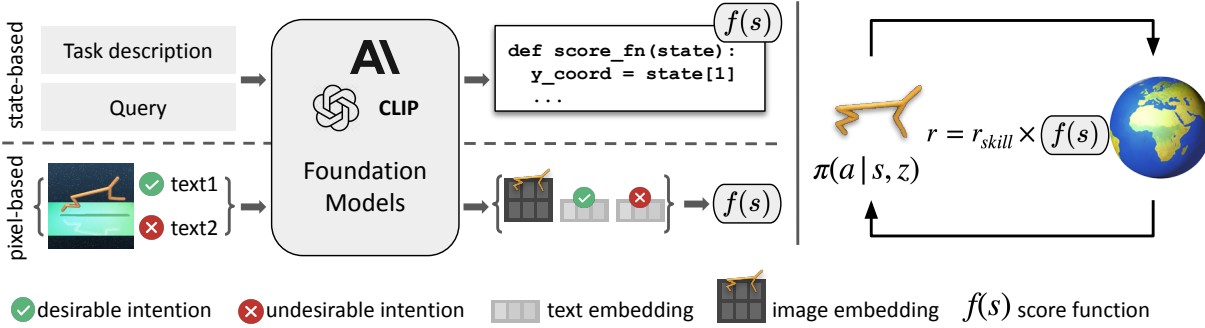

Figure 1: FoG leverages foundation models (such as ChatGPT, Claude and CLIP) to score states in relation to given commands during training. These scores are used to re-weight the rewards of the underlying skill discovery algorithm. **Left**: In state-based tasks (top row), task descriptions are provided to foundation models, which are queried to generate a score function $f(s)$ based on our requirements. In pixel-based tasks (bottom row), the current visual state, textual descriptions of desirable and undesirable intentions are input to foundation models to obtain embeddings. These embeddings are then used to form the score function $f(s)$, see Equation (9). **Right**: During training, rewards of the underlying skill discovery method ($r_{skill}$) are re-weighted using the score function. Re-weighting $r_{skill}$ (we use METRA (Park et al., 2023b)) by the score function is equivalent with using the score function as the distance metric in the DSD objective.

1. We propose FoG, a novel foundation model guided unsupervised skill discovery method that learns diverse and desirable skills.

2. We evaluate FoG alongside six state-of-the-art baselines on both state-based (i.e. structured, low-dimensional representations) and pixel-based tasks. FoG outperforms baselines in both scenarios, showcasing superior input-agnostic generalization capabilities.

3. We show FoG can learn behaviors that are challenging to define, such as being 'twisted' and 'stretched' on a humanoid robot, suggesting its potential for more complex applications.

4. We perform an extensive ablation study to assess the contribution of each component to FoG's performance.

## 2 Preliminaries and Problem Setting

We consider a reward-free Markov Decision Process defined as $\mathcal{M} = (\mathcal{A}, \mathcal{S}, p)$. $\mathcal{S}$ denotes the state space, $\mathcal{A}$ denotes the action space and $p$ is the transition dynamics function. A latent vector $z \in \mathcal{Z}$ (also called 'skill') is sampled during training and its conditioned policy $\pi(\cdot|s, z)$ is executed to get a skill trajectory $\tau = (s_0, s_1, ..., s_T)$ following the process: $p(\tau|z) = p(s_0) \prod_{t=0}^{T-1} \pi(a_t|s_t, z)p(s_{t+1}|s_t, a_t)$. $\pi(\cdot|s, z)$ can be learned by optimizing unsupervised exploration objectives we discuss below (distance-maximization) or in Section 5 (mutual information). The policy network $\pi(\cdot|s, z)$ takes the concatenation of the state observation $s$ and the skill vector $z$ as its input.

FoG utilizes the Distance-maximizing Skill Discovery (DSD) (Park et al., 2023a) objective. Unlike mutual information based methods (Eysenbach et al., 2018), DSD aims to maximize the Wasserstein dependency measure (WDM) (Ozair et al., 2019) defined as:

$$I_{\mathcal{W}}(S; Z) = \mathcal{W}(p(s, z), p(s)p(z)), \tag{1}$$

where $\mathcal{W}$ is the 1-Wasserstein distance on the metric space $(S \times Z, d)$ for distance metric $d$. By maximizing the objective in Equation (1), the agent will not only maximize the diversity of skills, but also maximize the distance metric $d$ (Park et al., 2023b). Under some simplifying assumptions (Ozair et al., 2019; Villani

et al., 2009), maximization of Equation (1) can then be rewritten as:

$$\sup_{\pi,\phi} \mathbb{E}_{p(\tau,z)} \left[ \sum_{t=0}^{T-1} (\phi(s') - \phi(s))^\top z \right] \quad \text{s.t. } \|\phi(x) - \phi(y)\|_2 \leq d(x,y), \ \ \forall(x,y) \in S, \tag{2}$$

where $\phi$ is a representation function that maps states to a $D$-dimensional space, which is the same as the skill space $Z$. Intuitively, Equation (2) aims to align the direction of $z$ and $\phi(s') - \phi(s)$ (to learn distinguishable and diverse skills), while maximizing the length of $\|\phi(s') - \phi(s)\|$, which leads to an increase in the distance between states based on the given distance metric $d$ due to the Lipschitz constraint (Park et al., 2023a). In principle, $d(x,y)$ in Equation (2) can be replaced by any of the distance metrics in Table 1, resulting in different unsupervised skill discovery methods. Equation (2) can be optimized with dual gradient descent, incorporating a Lagrange multiplier $\lambda$ and a small slack variable $\epsilon > 0$:

Update $\phi$ to maximize: $\qquad\qquad \mathbb{E}[(\phi(s') - \phi(s))^\top z] + \lambda \cdot \min(\epsilon, d(s,s') - \|\phi(s) - \phi(s')\|)$ (3)

Update $\lambda$ to minimize: $\qquad\qquad\qquad\qquad \lambda \cdot \mathbb{E}[\min(\epsilon, d(s,s') - \|\phi(s) - \phi(s')\|)]$ (4)

Update $\pi$ with reward: $\qquad\qquad\qquad\qquad\qquad\qquad\qquad (\phi(s') - \phi(s))^\top z$ (5)

For derivation of these equations we refer to Park et al. (2022; 2023a;b).

## 3 Foundation Model Guided Skill Discovery

FoG extracts a score function from foundation models based on human intentions to re-weight skill discovery rewards, illustrated in Figure 1. For state-based tasks, the foundation model is queried to output a score function aligned with our intentions. In pixel-based tasks, the score function is formed using state and intentional text embeddings from the foundation models. The skill-conditioned policy is then trained to maximize these re-weighted rewards during unsupervised skill discovery.

### 3.1 Score Function

We extract a score function from foundation models that can assign higher values for desirable state transitions and lower values for undesirable state transitions with respect to the given intentions. This score function is then used to re-weight rewards of the underlying skill discovery method. We define the score function $f : (S, S') \rightarrow [0, 1]$ which takes a state pair (current state and next state) as input and outputs a value between 0 and 1, indicating the desirability of the given state transition. This score function is then used to reweight the skill discovery rewards. The skill discovery reward $r_{skill}$ of Equation (5) therefore becomes:

$$r = f(s, s') \times r_{skill} = f(s, s')(\phi(s') - \phi(s))^\top z. \tag{6}$$

Since we use METRA (Park et al., 2023b) as the underlying skill discovery algorithm, using the score function to re-weight the METRA rewards is equivalent to using it as the distance metric in the DSD objective:

$$\sup_{\pi,\phi} \mathbb{E}_{p(\tau,z)} \left[ \sum_{t=0}^{T-1} (\phi(s_{t+1}) - \phi(s_t))^\top z \right] \quad \text{s.t. } \|\phi(s) - \phi(s')\|_2 \leq f(s, s'), \ \ \forall(s, s') \in S_{adj}, \tag{7}$$

where $S_{adj}$ represents the set of adjacent state pairs. The derivation of Equation (7) can be found in Appendix A. By using the score function as the distance metric in the DSD objective, FoG not only maximizes the diversity of skills, but also maximizes the output of the score function, leading to skills that are more aligned with our intentions.

In practice, we find that a binary score function works well, i.e. outputting 1 if the state transition is desirable and $\alpha$ if it is not, where $0 \leq \alpha < 1$. We examine different values of $\alpha$ and a non-binary score function in Section 4.

### 3.2 Implementation Details

We show that the score function $f(s, s')$ can be simplified to $f(s')$, which only depends on the next state $s'$. Let's consider the desirability of the current state $s$ and the next state $s'$ across two scenarios:

- **$s'$ is desirable:** Regardless of $s$, $f(s, s')$ should be high to encourage maintaining a desirable state (if $s$ is desirable) or transitioning from an undesirable one (if $s$ is undesirable).

- **$s'$ is undesirable:** Regardless of $s$, $f(s, s')$ must be low to discourage stagnating in an undesirable state (if $s$ is undesirable) or deviating from a desirable one (if $s$ is desirable).

Since the score depends solely on the next state, we simplify the function as:

$$f(s, s') = \begin{cases} \text{High} & \text{if } s' \text{ is desirable} \\ \text{Low} & \text{if } s' \text{ is undesirable} \end{cases} \implies f(s'). \tag{8}$$

Our work builds on top of METRA (Park et al., 2023b), which is the state-of-the-art unsupervised skill discovery method that works for both state-based and pixel-based input. FoG re-weights the skill discovery reward of METRA by the score function that is extracted from foundation models. For state-based tasks, we ask foundation models to generate the score function directly. For pixel-based tasks, we use foundation models to output embeddings to form the score function. All code is available through the supplemental materials.

**State-based** We ask ChatGPT or Claude to generate a score function $f(s)$ that equals 1 if the state satisfies our intentions, and $\alpha$ otherwise. Unlike Eureka (Ma et al., 2023), which queries foundation models to generate a reward function for training agents from scratch, FoG instead asks for a score function to modulate skill discovery. Prompt details for state-based tasks and examples of resulting output score functions are provided in Appendix F.7.1.

**Pixel-based** We use CLIP (Radford et al., 2021), a vision-language model that is trained to align images and text, to first generate embeddings for images (pixel-based states) and texts (textual descriptions of our intentions). Then, the score function is formed by computing the *Cosine* similarity between the image and text embedding. If the current state is more similar to the description of the desirable intention, the output is 1. Conversely, if it is more similar to the undesirable one, the output is $\alpha$. The score function can be expressed as Equation (9).

$$f(s) = \begin{cases} 1, & \text{if } Cosine(E_s, E_{t1}) > Cosine(E_s, E_{t2}). \\ \alpha, & \text{otherwise.} \end{cases} \tag{9}$$

where $E_s$ is the embedding of the current pixel-based state, $E_{t1}$ and $E_{t2}$ are the embeddings of the textual descriptions of desirable and undesirable intentions, respectively. Setting $\alpha = 0$ attempts to not learn undesirable behaviors at all (since $\alpha \times r_{skill} = 0$) while setting $\alpha = 1$ reduces FoG to the underlying skill discovery algorithm METRA. We examine different values of $\alpha$ in Section 4.3. Details of textual descriptions of desirable and undesirable intentions can be found in Appendix F.7.2.

## 4 Experiments

Our experiments aim to address the following questions:

- How does FoG perform in state-based tasks where more context and informative features are provided?

- In pixel-based tasks, where only visual information is provided, can FoG guide agents to learn diverse and desirable behaviors and skills?

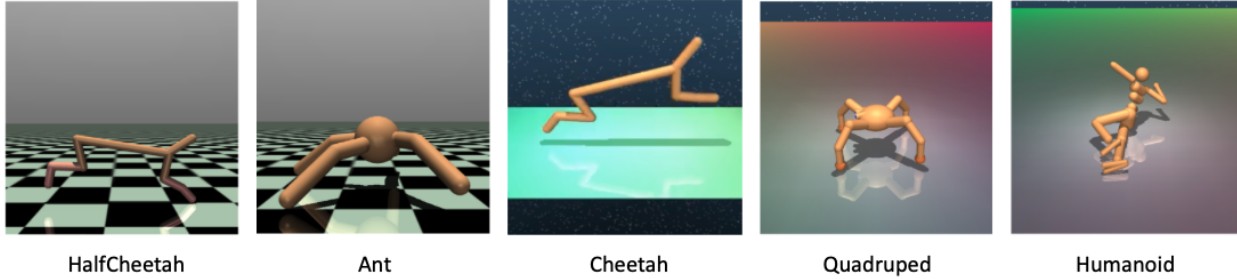

Figure 2: Environments used in our work. HalfCheetah and Ant are state-based while the other three are pixel-based.

We use common environments in unsupervised skill discovery literature, see Figure 2, including two state-based tasks and three pixel-based tasks: HalfCheetah and Ant are state-based tasks from OpenAI gym (Brockman et al., 2016), Cheetah, Quadruped and Humanoid are pixel-based tasks from DMC (Tunyasuvunakool et al., 2020).

We have seven baselines for FoG to compare against:

- METRA (Park et al., 2023b), the state-of-the-art unsupervised skill discovery method.

- METRA+, which integrates human intentions through hand-coded reward functions, and was also used as a baseline in DoDont (Kim et al., 2024b).

- LSD (Park et al., 2022), an unsupervised skill discovery method that maximizes DSD objective with Euclidean distance as the distance metric.

- DoDont (Kim et al., 2024b), a demonstration-guided unsupervised skill discovery method, learns diverse and desirable behaviors shown in the demonstrations. In some cases, it needs additional state-based inputs alongside with pixel-based input to work properly, more details can be found in Appendix D.

- DoDont+, a variant of DoDont that replaces expert demonstrations with demonstrations annotated using foundation models.

- FR-SAC, a SAC (Haarnoja et al., 2018) agent rewarded using scores from foundation models (**F**oundation **R**ewards) using Equation (14).

- METRA+FR, an agent that integrates FR into METRA by adding these two rewards together directly. Instead of using multiplication form in Equation (6), it uses the additive form.

All agents in the same task are trained with the same number of environment steps and all experiments are performed multiple times with different independent seeds (3 seeds in state-based and 8 seeds in pixel-based tasks), and average results with error bars are reported. For simplicity, we set $\alpha = 0$ for all experiments. Details about environments and baseline implementations can be found in Appendix F. See website[1] for videos of the learned behaviors and skills.

## 4.1 State-based Tasks

To test whether FoG can work in state-based tasks, we train FoG in HalfCheetah and Ant. Following the details in Section 3.2, we input the description of the tasks, information about state space and action space to foundation models as context, then ask foundation models to generate a score function that returns 1 when the requirement in the query is satisfied otherwise $\alpha$. In HalfCheetah, we train FoG to eliminate dangerous behaviors (flipping over). In Ant, we train FoG to avoid a specific area, in this case to not go south.

---

[1]https://sites.google.com/view/submission-fog

Results of these experiments are visualized in Figure 3, with generated score functions for both tasks on the top. We first of all see that foundation models can recognize feature dimensions of the state that are important for meeting our requirements. For example, in HalfCheetah the second dimension of the state is the angle of Cheetah's front tip, which is important for determining if the agent flips over or not. In Ant, the first dimension of the state is the y-coordinate of Ant, which can be used to locate the agent in a south-north position. We see foundation models clearly set the right threshold and implement the logic to fulfil the intention we asked for, i.e., if the angle of the Cheetah's front tip is larger than 1.57 in radians (90 degrees) it flips over, and if the y-coordinate of Ant is larger than 0 it is in the north part of the plane. By re-weighting the skill discovery rewards using the generated score function from foundation models, FoG learns to not roll in HalfCheetah while METRA flips a lot (two bottom-left sub-figures of Figure 3). In Ant, FoG learns to always move to north and METRA learns to go in every direction (bottom-right part in Figure 3). However, naively adding rewards of METRA and FR (METRA+FR) leads to static behaviors near the starting point without any movement.

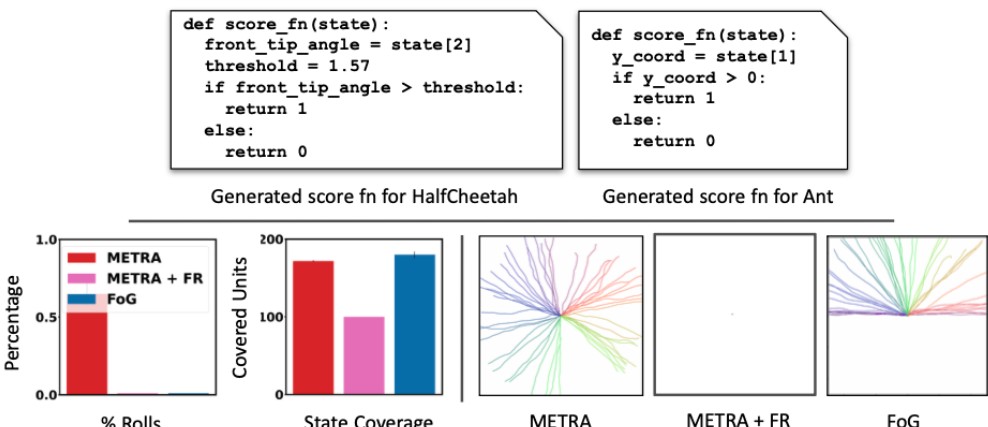

Figure 3: Comparison between METRA, METRA+FR and FoG on state-based HalfCheetah and Ant. In both tasks, foundation models successfully capture the relevant state dimension and set threshold for it. **Left**: FoG learns not to roll in HalfCheetah, while METRA rolls over 50% of the time, violating our intention. **Right**: FoG learns to not move to south in Ant, and METRA learns to move in all directions.

## 4.2  Pixel-based Tasks

We now conduct experiments in pixel-based tasks, where only visual information is available. Unlike in state-based tasks, where we ask foundation models to directly generate a score function, in pixel-based tasks we leverage foundation models to output embeddings of 1) the visual state and 2) textual descriptions of our desirable and undesirable intentions. The score function is then computed from Equation (9). We examine FoG in four aspects:

- Can FoG learn diverse skills while eliminating undesirable behaviors?

- Can FoG learn diverse skills without entering certain areas?

- Can FoG learn complex behaviors that are difficult to clearly define?

- What are the most critical design choices of FoG?

**Learn to eliminate undesirable behaviors**  We first focus on guiding the agent to learn desirable low-level behaviors (e.g., standing normal) while eliminating undesirable ones (e.g., flipping over) that could potentially damage the robot. In pixel-based Cheetah, we use 'agent flips over' and 'agent stands normally' as textual descriptions to express our intentions.

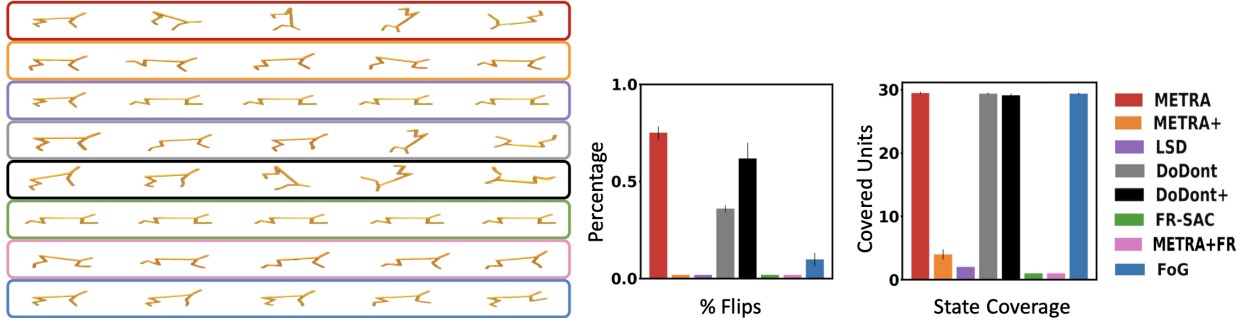

Figure 4: **Left**: Executions of example skills from different agents in pixel-based environment, Cheetah. From top to bottom: METRA, METRA+, LSD, DoDont, DoDont+, FR-SAC, METRA+FR, FoG. **Right**: Percentage of flips (which should be prevented based on the guidance) and state coverage for different agents. METRA, METRA+, DoDont, and DoDont+ discover diverse states but often flip. LSD and FR-SAC fail to learn diverse skills. FoG excels with high state coverage and minimal flipping.

As shown in the left part of Figure 4, FoG (the bottom one) consistently learns to run without flipping, demonstrating the lowest percentage of flips during evaluation. In contrast, other methods struggle to prevent flipping effectively. METRA flips in over 70% of episodes, DoDont in more than 35%, and DoDont+ in 50% of the episodes. LSD, FR-SAC, METRA+ and METRA+FR struggle to learn to move in different directions, discovering static behaviors and rarely flipping. Although METRA, DoDont, DoDont+ and FoG achieve similar state coverage, FoG effectively prevents flipping.

The poor performance of METRA+ suggests that defining a proper score function manually is not trivial (we follow the definition in Kim et al. (2024b) and use $r_{run} - r_{flip}$ as the score function). The poor performance of DoDont stems from the inaccurate classifier, which exploits the color of the ground to distinguish different states (normal and flipping postures), outputting high scores for unseen undesirable behaviors. A more in-depth analysis on the failure of DoDont can be found in Appendix D. FR-SAC fails to learn meaningful behaviors, suggesting only using foundation model scores to train RL agents is insufficient (see more analysis in Appendix E.4). To evaluate how these learned skills perform in downstream tasks, we train a controller to select from the learned set of skills. This controller trained using FoG skills shows quick adaptations in the downstream tasks, as shown in Appendix C.

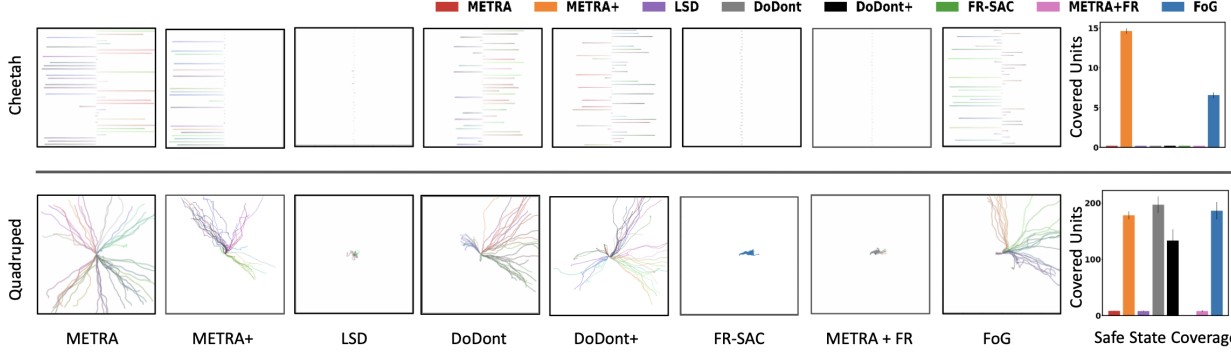

Figure 5: **Top**: Results on the pixel-based environment Cheetah, with learned skills shown in x-coordinates. METRA+ learns to perfectly avoid the undesirable area and FoG has a strong preference to go to the desirable area, as also clearly visible from the Safe State Coverage on the right. Other agents fail. **Bottom**: Results on the pixel-based environment Quadruped, with learned skills shown as xy-coordinates. Similar conclusions can be drawn regarding most of agents. Unlike in Cheetah, DoDont successfully learns to avoid the bottom-left areas.

**Learn to avoid hazardous areas**  Previous methods focus solely on maximizing skill diversity, often leading agents to explore in all possible directions. In practice, however, we want agents to avoid certain areas when they are hazardous. For instance, a robot operating in a factory should be able to avoid prohibited areas. To test whether FoG can learn to avoid certain areas (high-level policies, as opposed to low-level behaviors in Figure 4), we train FoG in the pixel-based versions of Cheetah and Quadruped. We designate the right area in Cheetah and the bottom-left area in Quadruped as hazardous and train agents to avoid them. Since there are no explicit indicators of directions in these two tasks, we express our intentions through colors. For example, in Cheetah, we use descriptions like '`ground is blue`' and '`ground is orange`' to signal whether the agent is on the left or right part and then form the score function following Equation (9).

Figure 5 illustrates the learned skills and 'Safe State Coverage' (the coverage of safe areas minus that of hazardous areas) of different agents. FoG clearly biases movement toward the safe areas. In Cheetah it prefers to go to the left part and in Quadruped it avoids the bottom-left area, resulting in higher safe state coverage than the baselines. In contrast, METRA explores all directions indiscriminately, LSD and FR-SAC fail to move, leading to the lowest safe state coverage. DoDont performs well in Quadruped but not in Cheetah (the classifier are unsure about initial states thus harm the exploration). The slightly worse performance of DoDont+ (compared to DoDont) in Quadruped stems from its inaccurate demonstrations annotated by foundation models. METRA+ performs the best, likely because that defining a score function in these tasks is straightforward (assigning 1 to states in safe regions and 0 for ones in hazardous regions (Kim et al., 2024b)). The results suggest that with expert-level demonstrations and 'perfect' hand-crafted score function, DoDont and METRA+ could potentially outperform FoG. However, the strength of FoG shines in scenarios where obtaining expert-level demonstrations or crafting a perfect score function is challenging, which is generally the case.

Non-expert demonstrations (like ones annotated by foundation models, which are used in DoDont+) introduce inaccuracies to the classifier, with annotation accuracy around 70%. This leads to an inaccurate classifier that consistently generates unreliable signals, ultimately resulting in poor performance. In contrast, FoG leverages CLIP on-the-fly. Although CLIP does not achieve perfect accuracy, it still allows the agent to learn effectively. As shown in Figure 7, the more accurate the scoring function, the better the performance of FoG.

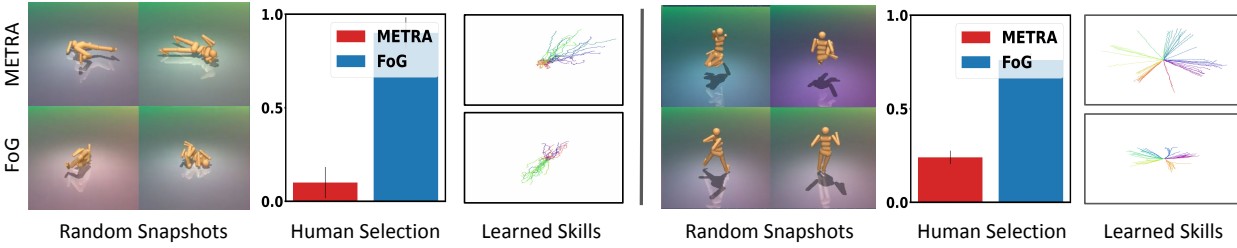

Figure 6: Learning results of METRA and FoG on Humanoid (**Left**) and Puppet (**Right**). Humans participants pick FoG to be more desirable 90% and 70% of the time in two tasks. Learned skills (shown in xy-coordinates) of different agents.

**Learning in Humanoid**  Humanoid is a challenging high-dimensional control task with a 21-D action space. Defining postures of this humanoid robot could be both hard and subjective, e.g. when it is "twisted" or "stretched", "running" or "walking", etc. This also makes it hard to design a reward function that can guide the agent to learn such behaviors. Since FoG uses foundation models, it overcomes this problem by directly evaluating whether a given frame or state is desirable—assigning higher scores to configurations like "twisted," which we want to encourage. This allows FoG to recognize and reward subtle behaviors that are otherwise hard to specify explicitly. We could not compare FoG with DoDont (Kim et al., 2024b) as the original paper does not include results on Humanoid, probably because demonstrations of a humanoid robot are challenging to obtain (an issue we also encountered).

First, we train FoG in the Humanoid task using intention descriptions '`agent is stretched`' and '`agent is twisted`'. To quantitatively assess whether the agent has successfully learned to twist, we create a

questionnaire and ask ten human participants to evaluate videos of different agents, selecting the ones they perceive as more "twisted". Videos and the questionnaire can be found on the project website and details of the experimental setup can be found in Appendix F.

In the left part of Figure 6, it is clear that FoG learns to exhibits more "twisted" postures while METRA tends to appear more "stretched". The 'Human Selection' shows how participants perceive the trained skills, with 90% of the time participants selecting FoG as more "twisted", further validating the observed outcomes. Both FoG and METRA successfully learn to move in different directions, highlighting the diversity of the learned skills. FoG's ability to move in different directions with "twisted" postures suggests its potential to guide agents in discovering skills involving behaviors with subjective definitions.

To further analyze FoG, we modify the 'Humanoid' task to a 'Puppet' variant, where the humanoid is pulled by a string above the head, i.e. the humanoid always keeps upright. The details of Puppet environment can be found in Appendix F.1. Besides learning diverse skills, we also ask the puppet to show running postures. See results in the right part of Figure 6. METRA learns to wriggle to all different directions with squat postures, whereas FoG learns to show more natural postures while moving in all directions. Similar to the Humanoid experiment, 70% of participants judged FoG to exhibit a more 'running' posture. See the website for videos.

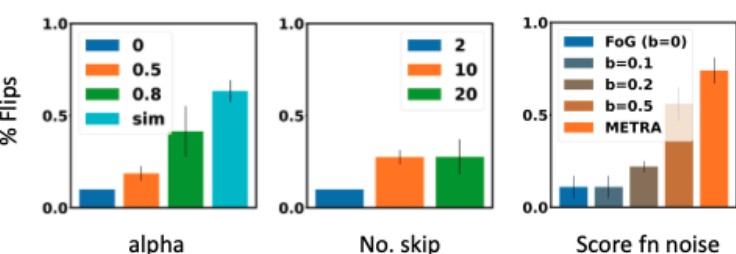

Figure 7: Percentages of flips that different FoG variants show on the Cheetah environment. Smaller $\alpha$, $N$ and $b$ return better performance.

### 4.3 Ablation Study

FoG introduces two hyperparameters. The first, $\alpha$ in the binary score function of Equation (9), controls the re-weighting of skill discovery rewards for undesirable states. Higher values make rewards for undesirable and desirable states less distinguishable, increasing the likelihood of agents learning undesirable behaviors. We evaluate three values, $\alpha = 0, 0.5, 0.8$. As shown in the left part of Figure 7, higher $\alpha$ leads to more undesirable behaviors (e.g., increased flipping in the Cheetah task). Directly using similarity of visual states and textual intentions (**sim**, calculated with Equation (14)) to re-weight rewards yields poor performance. While $\alpha = 0$ works well across experiments, it may overly constrain exploration in some cases (see examples in Appendix E.1).

In pixel-based tasks, obtaining embeddings for every pixel state is computationally expensive. Instead, embeddings are computed every $N$th state, with the score applied to the following $(N-1)$ states. Smaller $N$ values improve accuracy but increase costs. As shown in the middle part of Figure 7, smaller $N$ leads to fewer flips (better performance), but there is no significant difference between $N = 10$ and $N = 20$, suggesting behaviors in Cheetah are quite smooth thus skipping 10 or 20 states leads to similar results.

**Prompt robustness:** The FoG score function relies on textual intentions. Here, we examine the sensitivity of FoG in the pixel-based Cheetah, with using different undesirable textual prompts. Although these texts have similar meanings, we observe significant variance in performance across different phrasings; notably, obscure terms (e.g., 'Upend') lead to poor

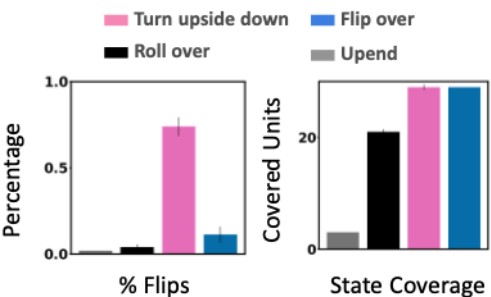

Figure 8: Performance of FoG on the Cheetah environment with different textual intentions used.

results. Our findings suggest that using common, straightforward descriptions such as 'Flip over' or 'Roll over' yields the best performance.

**Sensitivity to score function noise:** Although FoG's CLIP-based score function is not perfectly accurate, it still enables the agent to learn effective behaviors. To assess how performance depends on the score accuracy, we inject noise by flipping the score output ($0 \leftrightarrow 1$) with probability $b$ during training. As the score function becomes noisier, the percentage of flips in Cheetah increases (see Figure 7) while the state coverage remains mostly constant (all 29, except $26.7 \pm 0.67$ for $b = 0.5$). These findings indicate that while FoG is robust to some noise, improved scoring enhances performance.

## 5 Related Work

Mutual information (MI) based unsupervised skill discovery aims to maximize MI between latent skill variables and visited states to learn diverse and distinguishable skills. This line of research focuses on maximizing mutual information (MI) $I(\cdot; \cdot)$ between skills $Z$ and states $S$, i.e., $I(S; Z) = H(S) - H(S|Z) = H(Z) - H(Z|S)$, where $H(\cdot)$ denotes entropy. By associating states $s \in S$ with different latent skill vectors $z \in Z$, these methods learns diverse skills that are mutually distinct (Eysenbach et al., 2018; Sharma et al., 2019; Laskin et al., 2022). SASD (Kim et al., 2023) and EDL (Hussonnois et al., 2023) integrate preference into MI methods by a pre-defined function and human feedback, and they operate only with state-based input. In contrast, FoG eliminates the need for human involvement and supports both state and pixel-based input.

However, these MI-based methods do not always encourage the agent to discover distant states, as the MI objective can be satisfied by learning simple and static skills (Park et al., 2023b; 2022). To address this limitation, Park et al. (2023a) introduces a Distance-maximizing Skill Discovery (DSD) framework that learns diverse skills while maximizing the traveled distance under the given distance metric $d$. LSD (Park et al., 2022) uses Euclidean distance between states as the distance metric to encourage agents to visit states that are as far apart as possible. CSD (Park et al., 2023a) employs a density function over visited states as the distance metric, to encourage agents to visit less frequently visited states. However, LSD and CSD only work with state-based inputs and fail in pixel-based tasks. METRA (Park et al., 2023b) instead uses a temporal distance function that is applicable in visual tasks as well, as the distance metric to push the agent to discover states that are temporally far apart. LGSD (Rho et al., 2024) utilizes foundation models to first convert state-based inputs to text descriptions, then uses embedding distance between text descriptions as the distance metric to encourage agents to learn semantic diverse skills. DoDont (Kim et al., 2024b) employs demonstrations to guide agents in learning desirable behaviors. Specifically, it trains a classifier over the demonstrations of what the agent should and should not do, and uses it as a distance metric in DSD, encouraging agents to learn to maximize intentions of the given demonstrations. Some distance metrics used by different methods are summarized in Table 1. Note that FoG can be interpreted as using a score function extracted from foundation models as the distance metric in DSD. We refer to Section 3.1 for further details.

Table 1: Distance metrics used by different methods in the distance-maximizing skill discovery objective. $q_\theta$ is a density function parameterized by $\theta$. Temporal distance is defined as the minimum number of environmental steps needed for the agent to go from one state to another state. $s_{lang}$ is the textual description of the state $s$. $p_\varphi$ is a classifier parameterized by $\varphi$.

| LSD | CSD | METRA | LGSD | DoDont | Ours |
|---|---|---|---|---|---|
| $\lVert s' - s \rVert$ | $-\log q_\theta(s'\lvert s)$ | temporal dis | $\mathrm{dis}(s'_{lang}, s_{lang})$ | $p_\varphi(s', s)$ | score fn |

FoG is most closely related to DoDont and LGSD, as both these methods aim to incorporate human preferences into skill discovery. However, DoDont relies on expert demonstrations, which can be costly (Fu et al., 2024; Pertsch et al., 2021) or infeasible for tasks where human performance is limited (e.g., defining "stretched" or "twisted" posture for a humanoid robot). Additionally, DoDont's classifiers require ground-truth state-based inputs to avoid being misled by unrelated information when learning behavioral intentions (see examples in Appendix D). LGSD leverages language models but is limited to low dimensional state-based tasks. As the states (e.g. $[0.3, 0.2]$) need to be first converted to text (e.g. `object at position` $[0.3, 0.2]$)

with a rule-based annotator, then gpt4-turbo (Achiam et al., 2023) is queried to generated descriptions. Furthermore, querying gpt4-turbo in a step-wise, chat-style manner is expensive. In contrast, FoG utilizes vision-language models and extracts a score function, applied either once (state-based tasks) or via batch processing (pixel-based tasks), to re-weight the underlying skill discovery rewards. It therefore has a fast response time and works well in both state-based and pixel-based tasks. We discuss related work on using foundation models for RL in Appendix B.

## 6 Conclusion and Future Work

We propose a novel unsupervised skill discovery method, FoG, guided by foundation models to incorporate human intentions. FoG first extracts a score function from foundation models based on input intentions, assigning higher preference to desirable states and lower preference to undesirable ones. This score function is then used to re-weight the underlying skill discovery rewards. By optimizing re-weighted rewards, FoG discovers not only diverse but also desirable skills. In addition, we also show FoG can learn skills involving behaviors that are complex and subjectively defined.

FoG uses CLIP due to its widespread adoption, it is open-source with readily available pre-trained weights and lightweight to run (the variant we use, has only 0.4B parameters). Importantly, our framework is not tied to CLIP specifically: FoG is designed to work with any vision-language model that provides image-text embeddings which can be used to compute similarity scores (Equation (9)). We expect that stronger or task-specific vision-language models would further improve FoG's performance. We leave exploration of alternative backbones for future work.

Although FoG performs well, it is not without limitations. First, there is no guarantee that score functions generated by foundation models are always appropriate. Additionally, since the score function is defined based on individual states, FoG may struggle to capture process-based alignment. This limitation could be addressed by defining the score function over a sequence of states (Sontakke et al., 2024). Furthermore, we believe FoG could benefit from more advanced and task-specific foundation models (Liu et al., 2023a; Yao et al., 2024; Padalkar et al., 2023; Valevski et al., 2024). One could also explore the performance of FoG with more complex intentions and more challenging tasks. Some preliminary results can be found in Appendix E.3. We hope FoG inspires future efforts in incorporating human intentions in unsupervised skill discovery.

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

# Appendix

# Table of Contents

## A Derivation of Equation (7)

The original DSD objective is shown in Equation (2). It is crucial to define a appropriate distance metric to encourage agents to not only learn diverse skills but also maximize the given distance metric. Park et al. (2023b) uses the temporal distance as the distance metric for the DSD objective in METRA, shown in Equation (10).

$$\sup_{\pi,\phi} \mathbb{E}_{p(\tau,z)} \left[ \sum_{t=0}^{T-1} (\phi(s_{t+1}) - \phi(s_t))^\top z \right] \quad \text{s.t.} \ \ \|\phi(s) - \phi(s')\|_2 \leq 1, \ \ \forall(s,s') \in S_{adj}. \tag{10}$$

Now, we use the score function $f(s, s')$ to re-weight the METRA rewards to get the objective of FoG. The new objective (FoG) now becomes Equation (11):

$$\sup_{\pi,\phi} \mathbb{E}_{p(\tau,z)} \left[ \sum_{t=0}^{T-1} f(s, s') (\phi(s_{t+1}) - \phi(s_t))^\top z \right] \quad \text{s.t.} \ \ \|\phi(s) - \phi(s')\|_2 \leq 1, \ \ \forall(s,s') \in S_{adj}. \tag{11}$$

Following Kim et al. (2024a), let scaled state function $\tilde{\phi}(s) - \tilde{\phi}(s') = (\phi(s) - \phi(s'))f(s, s')$. By replacing $\phi(s) - \phi(s')$ with $(\tilde{\phi}(s) - \tilde{\phi}(s'))/f(s, s')$ and transforming the constraint in Equation (11) (since $f(s, s') \geq 0$), we derive Equation (12) (Equation (7)), which is using the score function as the distance metric in the DSD objective.

$$\sup_{\pi,\phi} \mathbb{E}_{p(\tau,z)} \left[ \sum_{t=0}^{T-1} \left( \tilde{\phi}(s_{t+1}) - \tilde{\phi}(s_t) \right)^\top z \right] \quad \text{s.t.} \ \|\tilde{\phi}(s) - \tilde{\phi}(s')\|_2 \leq f(s,s'), \ \forall(s,s') \in S_{adj}. \qquad (12)$$

Since we will optimize exclusively over this new scaled representation, for notational simplicity, we rename $\tilde{\phi}$ to $\phi$ in the final objective (Equation (7)). Hereby, we show that using the score function to re-weight the METRA rewards is equivalent as using it as the distance metric in the DSD objective.

## B  Extended Related Work

**Foundation Models in Reinforcement Learning** FoG leverages foundation models to guide unsupervised skill discovery in learning desirable behaviors. Thanks to success of foundation models (Touvron et al., 2023; Liu et al., 2023b) they can now be used to provide information for RL agents. Motif (Klissarov et al., 2023) and IGE (Lu et al., 2024) employs large language models to generate exploration bonuses. Eureka (Ma et al., 2023) uses large language models to generate reward functions for state-based robotic tasks, outperforming human designed reward functions across multiple tasks. Lift (Nam et al., 2023) uses LLM and VLM to guide learning in Minedojo (Fan et al., 2022). LAMP (Adeniji et al., 2023) and Rocamonde et al. (2023) utilize the similarity between pixel embedding and text-commands embedding, as output by a vision-language model, as the step-wise reward in visual robotic tasks. Results show that such step-wise signals alone barely work (matching the results we had

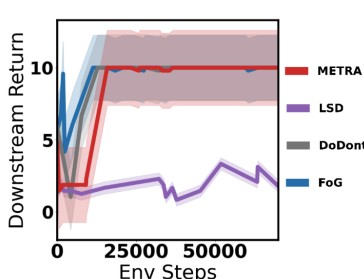

Figure 9: Downstream task performance.

in Section 4.3), and require either fine-tuning or special task modifications to perform well. Task-specific foundation models generally can achieve better performance on specific tasks, such as Minedojo (Fan et al., 2022) in Minecraft and EmbodiedGPT (Mu et al., 2024) in robotics. Despite this, FoG demonstrates that pre-trained foundation models, even without fine-tuning or any modifications of tasks, can be used to guide RL agents to discover diverse and desirable skills. In state-based tasks, FoG uses foundation models to generate a score function aligned with human intentions. Unlike Eureka (Ma et al., 2023), FoG: 1) avoids iterative feedback loops with the environment, as Eureka requires multiple rounds of feedback to refine the reward function, and 2) uses the score function to re-weight skill discovery rewards, whereas Eureka directly trains agents with the generated reward function.

## C  Downstream Tasks

After obtaining skills, we can train a controller to select these (frozen) learned skills to achieve given downstream goals. We follow the implementation of Park et al. (2023b), and set $g \sim [-10, 10]$ as the goal. During training, the agent receives a reward of 10 if the goal is reached. We train a controller to select a skill $z$ every $K = 50$ steps, and the learned policy $\pi(\cdot|s,z)$ is executed for $K$ steps. We use SAC (Haarnoja et al., 2018) for training the controller and all hyperparameters are kept the same as the METRA codebase. Results are shown in Figure 9. The controller that is trained using frozen skills learned by FoG shows better performance at the beginning and converges faster than the baselines, indicating that FoG effectively learns meaningful skills that can be quickly adapted

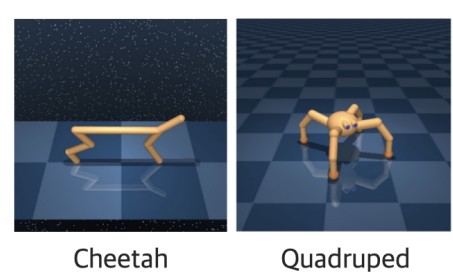

Figure 10: Tasks with non-colored ground that DoDont uses.

to downstream tasks. LSD does not learn useful skills thus the trained controller performs poorly. METRA slightly lags behind of DoDont.

## D   Analysis of DoDont

The performance of DoDont in our paper is quite different to the one from the original paper due to different experimental setup. Here, we provide a more in-depth analysis of why DoDont fails in our experiments.

**Failure in Figure 4**   To keep a fair comparison, we use pixel inputs for both the classifier and the RL part in DoDont (since FoG does not require ground-truth state information and works with only pixel inputs), which differs from the original experiments in the DoDont paper. In the Appendix D.1 of DoDont paper, authors mentioned that DoDont uses state information as input for the RL agent (both the policy and the critic). The classifier might exploit the background color as a shortcut to distinguish between different states rather than observing the agent's embodiment, thus DoDont instead uses a non-colored ground (see in Figure 10). However, the backbone of DoDont, METRA, cannot learn diverse skills without the colored ground (since there will be no indication of directions). Thus, DoDont uses state-information for the RL part. During the training,

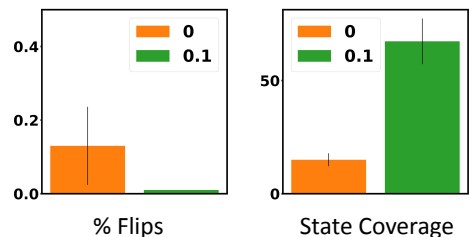

Figure 11: Results on the Quadruped task. Setting $\alpha = 0$ explores less (lower state coverage) thus results in worse performance (more flips).

image states are first input to the classifier to get rewards, then the corresponding compact ground-truth states are used to train the RL agents along with the rewards obtained from the classifier. Our experiments show that indeed, if pixel inputs are used for both the RL part and the classifier, DoDont fails (see results in Figure 4). The classifier indeed exploits the background color as a shortcut to distinguish between different states rather than observing the agent's embodiment, classifies unseen 'Dont' states as 'Do'. See videos on https://sites.google.com/view/submission-fog.

**Failure in Figure 5**   In Figure 5, DoDont successes in Quadruped while fails in Cheetah. The performance of the classifier shows that it is able to accurately classifying "going left" and "going right", but unsure about states at the beginning. Our intuition is that such uncertainty hurts the exploration at the beginning, resulting in poor performance later on. See videos on https://sites.google.com/view/submission-fog.

## E   Additional Experiments

### E.1   Quadruped Learns to Not Flip

Although we found that setting $\alpha = 0$ works well in experiments presented in Section Experiments, sometimes it might hurt the exploration. Similar with experiments performed in Figure 4, here, we train FoG to not flip in Quadruped. We see in Figure 11, FoG learns to not flip most of time (less than 20%) when setting $\alpha = 0$, but it almost always stays near the starting point and does not explore, resulting in low state coverage. After loosing $\alpha$ a bit and set it to 0.1, FoG learns to eliminate all flips and has a significant higher state coverage.

### E.2   Results on Franka Kitchen

To examine FoG in more complicated tasks, we train FoG in Franka Kitchen (introduced by Gupta et al. (2019)) with different textual descriptions of intentions, such as 'robotic arm is stretched', 'robotic arm is twisted' and 'robotic arm is on the right of the scene'. Results can be seen in Figure 12. By using different intentions, we see robotic arms clearly bias the movements to different areas. However, we did not find a way to use these skills to better solve the downstream tasks yet. We hope this could inspire future efforts in investigating FoG in more complex tasks.

We also found that the baseline method METRA learns to open the cabinet very often, e.g. in our case 78% of time. We test FoG to bias towards 'cabinet is closed'. It learns to stop opening the cabinet (only 22%), and please see the videos on our project website. We see the agent stops the intention to open the cabinet in most of cases.

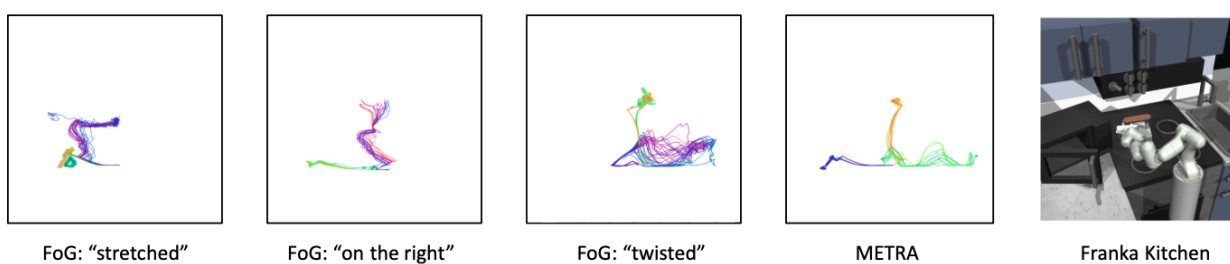

| FoG: "stretched" | FoG: "on the right" | FoG: "twisted" | METRA | Franka Kitchen |

Figure 12: In Franka Kitchen, different skills FoG learned with different textual descriptions of intentions. Skills are displayed with x-y coordinates of the robotic arm.

### E.3  Results on Multiple Intentions

In Section Experiments, only one intention is used in FoG. In principle, multiple intentions could be used simultaneously to form the score function. Then, Equation (9) becomes:

$$f(s) = \begin{cases} 1, & \text{if } Cosine(E_s, E_{t1}^1) > Cosine(E_s, E_{t2}^1) \text{ and} \\ & \quad Cosine(E_s, E_{t1}^2) > Cosine(E_s, E_{t2}^2) \text{ and} \\ & \quad ... \\ & \quad Cosine(E_s, E_{t1}^n) > Cosine(E_s, E_{t2}^n) \\ \alpha, & \text{otherwise.} \end{cases} \tag{13}$$

where $E_{t1}^n$ and $E_{t2}^n$ are the $n$th textual descriptions of our intentions.

Now, the score function $f(s)$ only assigns higher values to desirable states when all provided intentions are satisfied. For example, we could ask FoG to not only learns to not flip, but also to avoid the right area. The textual descriptions we should use are: 1) 'agent flips over', 'agent stands normally'; 2) 'ground is Yellow-Orange', 'ground is Green-Blue'. See the result in Figure 13, the agent does not learn to avoid the right part at all but it does learn to eliminate flips (not shown in the figure). We found that using multiple intentions restricts the exploration too much so that the agent might just learn to fulfill one intention and ignore others or ignore all of them and learns to not move at all. Using multiple intentions in FoG still needs more investigations and we hope the preliminary results and ideas presented in this section could inspire future efforts.

### E.4  Using scores as step-wise reward signals in FR-SAC

FoG uses foundation model scores to re-weight the unsupervised skill discovery rewards, learning diverse and desirable behaviors. However, directly optimizing these scores is not ideal. In Figure 14, scores for pre-collected episodes aligned with human intentions ('Yes') and misaligned ones ('No') reveal significant noise despite correct overall trends (we use the same textual intentions from previous experiments, i.e. Cheetah in Figure 4 and Quadruped in Figure 5). For example, in Cheetah, after flipping upside down at step 50, the agent consistently receives low scores. In Quadruped, scores either remain high or gradually decrease as the agent moves diagonally. This noise makes direct score optimization unreliable. As can be seen in Figure 4 and Figure 5, the agent trained solely with such noisy reward signals (FR-SAC) learns only static postures, resulting in low (safe) state coverages, suggesting that directly optimizing these scores is insufficient.

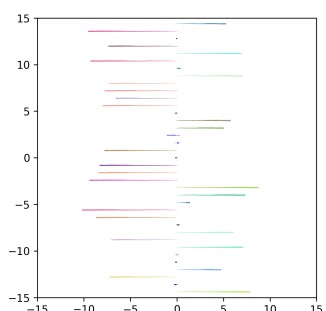

Figure 13: Skills learned by FoG with two intentions.

## F  Experiment Details

### F.1  Environment Details

**State-based**  HalfCheetah and Ant are from OpenAI Gym (Brockman et al., 2016). The state space of HalfCheetah is 18-dimensional and the one of Ant is 29-dimensional. HalfCheetah has a 6-dimensional action space while Ant has a 8-dimensional action space.

**Pixel-based**  Cheetah, Quadruped and Humanoid are from DeepMind Control Suite (Tunyasuvunakool et al., 2020). Following previous work (Lee et al., 2021; Park et al., 2024; 2023b), pixel-based DMC tasks are all with gradient-colored floors to indicate different directions. The size of visual observations is $64 \times 64 \times 3$. The dimension of action space for Cheetah, Quadruped and Humanoid are 6, 12 and 21, respectively. The episode length is 200 for Ant, HalfCheetah and Cheetah, 400 for Quadruped and Humanoid.

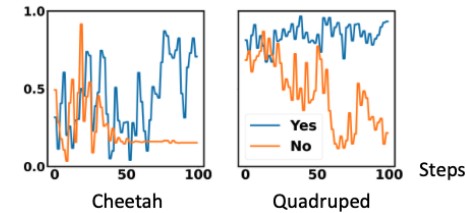

Figure 14: Scores outputted by foundation models on the pre-collected episode that are (not) aligned with human intentions.

**Modified Humanoid**  Since none of existing unsupervised skill discovery methods can train the visual Humanoid agent to stand up, limiting FoG to showcase more interesting behaviors, such as running, etc. We created a 'Puppet' task based on the DMC Humanoid environment, see Figure 15. The humanoid robot is pulled by a puppet anchor on the top of its head. Thus, the humanoid robot keeps standing by default and never falls down. The anchor also moves with the humanoid.

### F.2  Baseline Details

**METRA**  We take the official codebase[2] from Park et al. (2023b) and use default hyperparameters for all experiments performed in this paper.

**METRA+**  We follow the implementation of METRA+ in the DoDont paper. For experiments in Figure 4, we use $r_{run} - r_{flip}$ as the reward. For experiments in Figure 5, we assign $+1$ for the safe region and 0 for the hazardous region.

**LSD**  We take the codebase of METRA, by setting correct arguments (turning off the dual regularization and turning on the spectral normalization), to run LSD. Detailed instructions can be found in the METRA codebase.

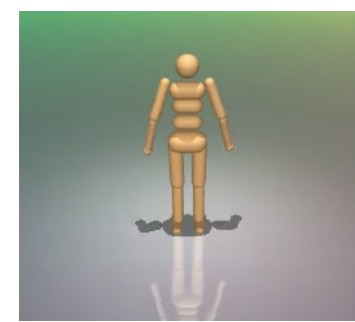

Figure 15: The Puppet environment.

**DoDont**  We take the official codebase from Kim et al. (2024b) and implement the training of the instruction net ourselves. We use eight demonstrations for each task, so four for "dos" and four for "donts". Demonstrations are obtained from trained FoG agents and can be found on our project website: `https://sites.google.com/view/submission-fog`. We stop the training of the classifier after it has more than 97% of accuracy.

**DoDont+**  A variant of DoDont, instead of using expert-level demonstrations, it uses demonstrations annotated by foundation models. In our case, we use CLIP to score frames (follow Equation (9)) in demonstrations that are used to train DoDont, and assign frames with score of 0 in the "dos" demonstration to "donts" demonstrations, and vice versa. Since CLIP cannot perfectly score frames, some states from "dos" demonstration are moved to "donts" demonstrations, and some states from "donts" demonstration are moved to "dos" demonstration. After training, the classifier of DoDont+ has about 70% of accuracy.

---

[2]`https://github.com/seohongpark/METRA`

**FR-SAC**   A soft actor-critic RL agent with using the score function as the reward function. We reuse the FoG codebase and set the number of skills to 1. Then, we train the skill-conditioned policy with the scores obtained from the foundation model (i.e. using the score function as the reward function), reducing to a normal RL agent.

### F.2.1   Hyperparameters Details

We use $\alpha = 0$ and $N = 2$ for all our experiments, unless otherwise mentioned. We train all agents in the same task with the same number of epochs and the performance at the end of training is reported. Details can be seen in Table 2. The same number of episodes is executed in each epoch, and within each episode the same number of environment steps is taken. We train continuous skills and the number of dimensions we used to train all agents in each task can be found in Table 2. We refer readers to read Park et al. (2023b) for details of all used hyperparameters.

Table 2: Number of epochs and dimensions of skills we used for training agents in different environments.

| HalfCheetah | Ant | Cheetah | Quadruped | Humanoid |
|---|---|---|---|---|
| 9000 | 9000 | 2000 | 3000 | 4000 |
| 4D | 2D | 4D | 4D | 2D |

### F.3   Non-binary Score Function

Instead of using a binary score function in Equation (9), we can also form a non-binary score function.

$$f(s) = \frac{e^{Cosine(E_s, E_{text1})}}{e^{Cosine(E_s, E_{text1})} + e^{Cosine(E_s, E_{text2})}}, \tag{14}$$

where $E_s$ is the embedding of the current pixel-based state, $E_{text1}$ is the embedding of textual descriptions of the desirable intention and $E_{text2}$ is the embedding of textual descriptions of the undesirable intention.

### F.4   Computation Usage

We run our experiments on an internal cluster consisting of A100 and H100 GPUs. Each run takes no more than 24 hours. Since FoG utilizes foundation models, it incurs additional computational overhead, approximately 1.5x slower than METRA.

### F.5   Experimental Setup for Human Judge

In Figure 6, we train FoG to be twisted in DMC Humanoid task. However, it is difficult for human to design a reward function to measure if learned skills contains more twisted postures or not. Thus, we ask human to be the judge to tell if FoG learns more twisted skills than the ones learned by the baselines.

We pick ten skills of each method randomly, in this case, FoG and METRA, and then pair them randomly. Participates are asked to select the video that shows the most 'twisted' behaviors without given any other information. Please see the full questionnaire we used in `https://sites.google.com/view/iclr-fog/questionnaire-of-humanoid`.

### F.6   Foundation Models

For state-based tasks, we query ChatGPT[3] or Cluade[4] to generate score functions that meet our requirements. For pixel-based tasks, we use pre-trained CLIP (clip-vit-large-patch14) from huggingface[5].

---

[3]`https://chatgpt.com`
[4]`https://claude.ai/new`
[5]`https://huggingface.co/openai/clip-vit-large-patch14`

### F.7  Prompts Used

### F.7.1  State-based Tasks

**Input to foundation models for HalfCheetah** [Descriptions of the task, state space and action space] According to the given info, could you please write a python function to check if the cheetah is flipped over or not. If yes, output 1 otherwise 0. **Output:**

```python
import numpy as np

def is_cheetah_flipped(observation):
    """
    Check if the HalfCheetah is flipped over.

    Args:
    observation (np.array): Observation array from the environment.

    Returns:
    int: 1 if the cheetah is flipped over, 0 otherwise.
    """
    # The second element in the observation array is the angle of the front tip
    angle = observation[1]

    # Define the threshold for flipping (e.g., |angle| > π/2)
    threshold = np.pi / 2

    if abs(angle) > threshold:
        return 1
    else:
        return 0

# Example usage:
# Assume `obs` is the current observation from the environment
# obs = env.reset()
# print(is_cheetah_flipped(obs))
```

**Input to foundation models for Ant** [Descriptions of the task, state space and action space] The agent always starts at [0, 0]. According to the given info, could you please write a python function to check if the agent is in the north part or not. If yes, output 1 otherwise 0. **Output:**

```python
def is_in_north(state):
    """
    Check if the agent is in the northern part (y > 0).

    Parameters:
    state (list or numpy array): The state space observation vector. The y-posit

    Returns:
    int: 1 if the agent is in the northern part, 0 otherwise.
    """
    # Assuming the second element in the state space corresponds to the y-coordi
    y_position = state[1]

    # Check if the y-coordinate is greater than 0
    if y_position > 0:
        return 1
    else:
        return 0
```

### F.7.2 Pixel-based Tasks

Textual descriptions of intentions we used for Cheetah:

- Figure 4: `The simulated two-leg robot flips over`, `The simulated two-leg robot stands normally`

- FR-SAC agent in Figure 4: `The simulated two-leg robot flips over`, `The simulated two-leg robot is running normally`

- Figure 5: `The underneath plane is Yellow-Orange`, `The underneath plane is Green-Blue`

Textual descriptions of intentions we used for Quadruped in Figure 5: `The underneath plane is Pink-Purple`, `The underneath plane is Green-Blue`.

Textual descriptions of intentions we used for Humanoid in Figure 6: `The simulated humanoid robot is stretched`, `The simulated humanoid robot is twisted`.

