# OpenReview forum: "Guiding Skill Discovery with Foundation Models"
_TMLR — Rejected by TMLR_

### Review · Reviewer_AeWi · 2025-11-10

**Summary Of Contributions:**

1. The paper proposes a novel skill discovery algorithm called FoG, which leverages a foundation model to guide the learning process.
   * The user provides natural language descriptions of both desired and undesired behaviors.
   * The foundation model evaluates the similarity between the current state and these descriptions, computing a corresponding score.
   * Using this score, FoG re-weights the rewards within the skill discovery framework to encourage diverse yet goal-aligned behaviors.

2. The paper demonstrates that FoG can effectively learn skills that align with user intentions in both state-based and pixel-based environments.

3. The paper includes an ablation study that analyzes the individual contributions of each component in the proposed method.

**Audience:**

No

**Audience Explanation:**

I **do not recommend acceptance** of this paper in its current form, for the following reasons:

1. **Limited Algorithmic Novelty**
   The proposed method appears to share substantial similarity with existing approaches such as LGSD and DoDon’t, with relatively modest conceptual differences. If the authors wish to emphasize the introduction of a new “metric,” it would be important to provide a rigorous mathematical justification, at least demonstrating that the proposed function satisfies the properties of a (pseudo-)metric.

2. **Issues in Theoretical Derivation**
   The derivation supporting one of the core claims appears to contain errors, which lead to a conclusion that does not hold as stated. In particular, the re-weighting procedure does not seem to be equivalent to introducing a distance metric, contrary to the claim in the paper. This undermines the theoretical contribution as currently presented.

3. **Concerns with Experimental Evaluation**
   Some experimental setups raise concerns. In certain cases, the chosen baselines do not seem well aligned with the problem setting, and in others, the evaluation metrics or comparison protocol appear to be misconfigured. These issues make it difficult to draw strong conclusions from the reported empirical results.

**Broader Impact Concerns:**

I have no concern

**Claims And Evidence:**

No

**Claims Explanation:**

1. One of the main claims of the paper—that *“re-weighting the MERTA rewards is equivalent to using the weight as the distance metric in the DSD objective”*—appears to be incorrect. The authors provide a derivation in Appendix A; however, in Equation (10), the correct expression inside the square brackets should be $[f(s_{t+1})\phi(s_{t+1}) - f(s_t)\phi(s_t)]^\top z,$ not  $ f(s_{t+1})(\phi(s_{t+1}) - \phi(s_t))^\top z $. The omission of $\phi(s_t)$ in Equation (10) substantially changes the result. \
  \
  If Equation (10) were corrected accordingly, the resulting formula would no longer correspond to a valid distance metric. Intuitively, a distance metric must be a function of two states $s$ and $s'$; $f(s')$ alone cannot define such a metric. \
  \
  Consequently, if it is not a distance metric, the main contribution of the algorithm reduces to re-weighting the reward using the foundation model rather than introducing a fundamentally new metric-based formulation.

2. For state-based tasks, it is difficult to see the necessity of using a foundation model, as the resulting score function appears overly simplistic. The function shown in Section F.7.1 consists of only a few lines of code—essentially an if–else condition checking whether the front tip angle exceeds $\pi / 2$.
  Such a rule can be easily designed by a human without the help of a foundation model, which raises doubts about its necessity in this setting. It would strengthen the paper if the score function captured patterns or semantics that are difficult to specify manually.

3. The ablation study in Section 4.3 could be somewhat misleading. The paper argues that the scores provided by the foundation model are highly noisy, suggesting that directly using them as rewards is suboptimal. However, this concern appears to apply equally to FoG itself. If the foundation model signal is noisy, it is not immediately clear why re-weighting the diversity reward with this same signal should yield better performance than using it directly as a reward. Providing a theoretical explanation or empirical evidence for this difference would help clarify the contribution. \
  \
  Furthermore, the results seem inconsistent with the earlier claim regarding noise in the foundation model signal. Despite the claimed noise, FR-SAC manages to learn a stable behavior with no flips. In my view, the lack of diversity in FR-SAC is not necessarily a shortcoming, since the method does not incorporate a diversity reward term. A more meaningful baseline might combine the foundation-model-derived reward (FR) with the diversity term from METRA, to evaluate whether such a hybrid could produce behaviors that are both diverse and aligned with human intent. If the signal is indeed noisy, it remains unclear how FR-SAC is able to learn such a consistent, non-flipping behavior. \
  \

4. In Section 4.1, the chosen baseline appears relatively weak. Including a stronger comparison, such as LGSD (Rho et al., 2024), would provide a fairer and more convincing evaluation, since LGSD can also demonstrate comparable levels of behavioral diversity.
  \
5. For some experimental results, particularly Figures 8 and 9, appears that the experiments were not repeated multiple times, as the plots do not include shaded areas indicating variance. This omission weakens the statistical validity of the results, especially in Figure 9, where the differences between curves are small.

**Requested Changes:**

1. (critical) **Proof Correction** : Please revise the derivation in **Appendix A**. As mentioned earlier, there appears to be an error in the proof that affects the validity of the claim. Ensuring that the derivation is mathematically consistent would help clarify the theoretical foundation of the paper.

2. (optional) **Presentation Improvements**

   * In **Section 4.2**, the second bullet point seems to have a missing portion in the latter part of the sentence. Please review and complete the statement for clarity.
   * In **Figure 8**, please provide a label for the x-axis. It appears to represent training steps, but clarification would be helpful. Additionally, it seems that each curve may correspond to a single episode at different steps. If so, this figure reflects a single episode rather than aggregated results across multiple episodes, which contradicts the caption’s description (“pre-collected episodes”). Please clarify this discrepancy in the text or figure caption.

3. (critical) **Baselines for State-Based Tasks** : For the **state-based task**, please consider including stronger and more relevant baselines. Currently, the comparison is made only with unguided unsupervised skill discovery methods. Adding guided or semi-supervised baselines would provide a fairer and more informative evaluation of the proposed method.

4. (critical) **Right ablation study** :  For the ablation study, please compare the if baseline follows the human intent well, instead of comparing diversity against pure model-free SAC. Alternatively, if the goal is to evaluate diversity, consider using an additive formulation as a baseline, such as Foundation reward + METRA  reward.

---

> ### Author Response · Authors · 2025-12-17
> **Author Response Part 1 of 2 (17/12/25)**
>
> Thank you for your review! We first address all your requested changes and then go through all questions you had below:
>
> **Requested Changes**
>
> > Proof Correction : Please revise the derivation in Appendix A. As mentioned earlier, there appears to be an error in the proof that affects the validity of the claim. Ensuring that the derivation is mathematically consistent would help clarify the theoretical foundation of the paper.
>
> Thank you for pointing this out. We acknowledge that there were issues in the original equations, and we have revised them accordingly. Specifically, we made the following improvements:
>
> -----
>
> 1. Refined Derivation of Eq. 7. Please refer to updated Eq. 7 and Eq. 8, the newly added Eq. 9, and their revised derivations in the Appendix A.
>
> We have redefined $f(s)$ as $f(s, s')$, a score function measuring the desirability-based distance between states $s$ and $s'$. This function quantifies the favorability of a transition from $s$ to $s'$, as determined by text intentions, in comparison to undesirable intentions.
>
> Consequently, the expression in Eq. 11 (originally Eq. 10) now becomes: $f(s, s') (\phi(s_{t+1}) - \phi(s_t))$. In the revised derivation, we scale the following: $\tilde{\phi}(s) - \tilde{\phi}(s') = (\phi(s) - \phi(s')) f(s, s')$. Then by replacing $(\phi(s) - \phi(s')) f(s, s')$ with $\tilde{\phi}(s) - \tilde{\phi}(s')$ and transforming the constraint in Eq. 11, it leads to the final objective in Eq.12.
>
> -----
>
> 2. Proof of Equivalence: $f(s, s') = f(s')$. Please see the updated explanation in Section 3.2 and newly added Eq. 8.
>
> We demonstrate that $f(s, s')$ is equivalent to $f(s')$, ensuring our original experimental results remain valid. To justify this equivalence, we consider four transition cases:
>
> * Case 1: If both $s$ and $s'$ are desirable, $f(s, s')$ should be high.
> * Case 2: If $s$ is undesirable and $s'$ is desirable, $f(s, s')$ should be high, as we aim to encourage transitions toward desirable states.
> * Case 3: If $s$ is desirable and $s'$ is undesirable, $f(s, s')$ should be low, since we seek to prevent transitions to undesirable states.
> * Case 4: If both $s$ and $s'$ are undesirable, $f(s, s')$ should be low.
>
> These cases reduce to the scoring function that only depends on the next state $s'$: we assign a high score if the next state $s'$ is desirable (regardless of the current state $s$) and a low score otherwise. Thus, the score depends solely on the desirability of the next state, confirming that $f(s, s') = f(s')$.
>
>
> > Presentation Improvements
>
> Thanks for pointing these out. We completed the statement in Section 4.2 and added the label for the x-axis in Fig. 14 (originally Fig. 8, we moved it to appendix as suggested for a clearer ablation section).
>
> > Baselines for State-Based Tasks
>
> As suggested, we now added a METRA+FR baseline in state-based tasks (both HalfCheetah and Ant) for a more fair comparison. See the updated Fig.3 in the revised manuscript. In Cheetah, METRA+FR learns not to flip at all like FoG but has a way lower state coverage than FoG. In Ant, METRA+FR learns not to move at all (monotonic behaviors), and performs poorly on state coverage. Indicating directly adding FR with METRA reward leads to poor performance (which is also consistent with findings by Section 5.3.2 in DoDont[1], i.e. additive form performs poorer than multiplication form). Additionally, we also run METRA+FR on all pixel-based experiments. Please see updated Fig. 4 and 5 in the revised manuscript.
>
> > Right ablation study
>
> We have now improved the ablation by:
> * As you suggested, we now added one more baseline METRA+FR, which combines FR with METRA reward itself. See results in updated Fig. 3, Fig. 4 and Fig. 5.
> * Added experiments on prompt robustness for the pixel-based Cheetah, which examines the effect of different textual intentions in FoG (as suggested by Reviewer **B3Th**).
> * Moved the analysis of FR (originally Fig.8) to the appendix.
>
>
> -------
>
> [1] Kim, Hyunseung, et al. "Do's and Don'ts: Learning Desirable Skills with Instruction Videos."

---

> > ### Author Response · Authors · 2025-12-17
> > **Author Response Part 2 of 2 (17/12/25)**
> >
> > **Questions and Concerns**
> >
> > > Consequently, if it is not a distance metric, the main contribution of the algorithm reduces to re-weighting the reward using the foundation model rather than introducing a fundamentally new metric-based formulation.
> >
> > We acknowledge that there were issues in the original equations, and we have revised them accordingly. Please see the answer to the first point for Requested Changes above, also in the revised manuscript (Eq. 7 and Eq. 8, the newly added Eq. 9, and their revised derivations in the Appendix A).
> >
> > > For state-based tasks, it is difficult to see the necessity of using a foundation model, as the resulting score function appears overly simplistic. ...
> >
> > The goal of the state-based experiments in Section 4.1 is not to argue for the necessity of a foundation model in this setting, but to demonstrate that the proposed pipeline (using a score function to re-weight skill discovery) does work as intended in a simpler setting.
> >
> > Our main contribution and emphasis are on the pixel-based tasks in Section 4.2, where agents rely solely on visual observations. In these settings, manually designing score functions is significantly more difficult as we don’t have access to the underlying state and tasks are more difficult.
> >
> > > The ablation study in Section 4.3 could be somewhat misleading. ... However, this concern appears to apply equally to FoG itself. If the foundation model signal is noisy, it is not immediately clear why re-weighting the diversity reward with this same signal should yield better performance than using it directly as a reward. ...
> >
> > We agree that the noisy foundation model signal can also affect FoG. As shown in the ablation study in Fig. 7, directly using the similarity score to reweight the underlying METRA reward (denoted as *sim*) leads to poor performance, confirming that naively applying the noisy signal is suboptimal.
> >
> > In contrast, FoG applies a binarized score function rather than the raw similarity signal. This discretization mitigates the impact of noise and allows the score to act as a coarse desirability filter instead of a dense reward, resulting in improved performance.
> >
> > > Furthermore, the results seem inconsistent with the earlier claim regarding noise in the foundation model signal. ...
> >
> > FR-SAC indeed learns a stable but monotonic static posture, which is a typical case for incapable unsupervised skill discovery methods. Similar behavior is also observed in other limited baselines such as LSD and METRA+, which fail to produce diverse skills.
> >
> > As you suggested, we added an additional baseline that directly combines the foundation-model reward with the METRA diversity reward (denoted as METRA+FR) in both state-based and pixel-based settings (see updated Fig. 3–5). Consistent with prior work (e.g., DoDont[1]), this direct combination performs poorly, suggesting that simply adding a noisy foundation-model signal to a diversity objective is insufficient.
> >
> >
> > > In Section 4.1, the chosen baseline appears relatively weak. Including a stronger comparison, such as LGSD[1], would provide a fairer and more convincing evaluation, since LGSD can also demonstrate comparable levels of behavioral diversityˇ
> >
> > While LGSD is indeed relevant, it requires per-state querying and does not extend to pixel-based observations, which makes a direct comparison less fair and limits its applicability to our main setting.
> >
> > To strengthen the evaluation, we additionally include METRA+FR (as suggested above) in both state-based and pixel-based experiments.
> >
> > > For some experimental results, particularly Figures 8 and 9, appears that the experiments were not repeated multiple times, as the plots do not include shaded areas indicating variance. This omission weakens the statistical validity of the results, especially in Figure 9, where the differences between curves are small.
> >
> > Fig.8 aims to show how noisy the FR could be, which can already be observed from one episode of data. Now we also moved it to the appendix E.4 as suggested.
> >
> > We now also included 3 independent seeds for Fig.9. Please see updated Fig.9 in the revised manuscript.
> >
> > -------
> > [1] Kim, Hyunseung, et al. "Do's and Don'ts: Learning Desirable Skills with Instruction Videos."
> >
> > [2] Rho, Seungeun, et al. "Language guided skill discovery."

---

> ### Comment · Reviewer_AeWi · 2026-01-13
>
> Thank you for your response. However, my concern still remains.
>
> If the score function ( f(s, s') ) is intended to be used as a distance measure, it must satisfy the properties of a metric, or at least a pseudo-metric. One of the required conditions for a pseudo-metric is **symmetry**, i.e.,
> $$
> f(x, y) = f(y, x), \quad \forall (x, y) \in \mathcal{X}.$$
>
>
> However, based on your explanation, if the score satisfies $f(s, s') = f(s')$, then in general
> $$
> f(s, s') = f(s') \neq f(s) = f(s', s),
> $$
> which violates the symmetry condition. This shows that $f$ is not even a pseudo-metric.
>
> More broadly, it is not intuitive that an arbitrary foundation-model query $f(s, s')$ would satisfy the necessary metric properties. Prior work has explicitly addressed this issue: **METRA** formally proves that the temporal distance they use is a pseudo-metric, and **LGSD** similarly establishes that the language-based distance satisfies pseudo-metric properties.
>
> If this work proposes a novel distance function to be integrated into the DSD formulation, then establishing its (pseudo-)metric validity is essential. Since the proposed function does not satisfy these properties, this significantly undermines the central contribution of the paper.

---

> > ### Author Response · Authors · 2026-01-14
> > **Response to the symmetry concern**
> >
> > Thank you for this insightful question. We acknowledge that our score function $f(s')$ does not satisfy symmetry. However, the DSD framework does not require $d$ to be a valid distance metric.
> >
> > This was formally established in Theorem 4.1 of CSD [1], which proves that given any non-negative function $d: S \times S \rightarrow \mathbb{R}^+_0$, the DSD framework implicitly induces an equivalent valid pseudometric $\tilde{d}$. The CSD paper explicitly states (Section 4.1):
> >
> > > "DSD has the nice property that $d$ does not have to be a valid metric. This is because DSD implicitly converts the original constraint into one with a valid pseudometric $\tilde{d}$. As a result, we can use any arbitrary non-negative function $d$ for DSD."
> >
> > Since our score function $f(s') \in \{\alpha, 1\}$ with $0 \leq \alpha < 1$ is non-negative, it satisfies Theorem 4.1's precondition, and the DSD framework automatically induces a valid pseudometric.
> >
> > We note that several accepted works also use asymmetric, non-metric distance functions within the DSD framework:
> > - CSD [1] uses $d_{\text{CSD}}(s, s') \propto -\log q_\theta(s'|s)$, which is asymmetric
> > - DoDont [2] uses a learned classifier $p_\phi(s, s')$, which is also asymmetric
> >
> > We will clarify this theoretical property in the revised manuscript and add a reference to Theorem 4.1.
> >
> > We hope this addresses the reviewer's concern. Should any aspects remain unclear or if the reviewer has further suggestions on how to better present this theoretical justification, we would be very happy to discuss and incorporate the feedback.
> >
> > ---
> >
> > [1] Park, S., Lee, K., Lee, Y., & Abbeel, P. "Controllability-Aware Unsupervised Skill Discovery." ICML 2023
> >
> > [2] Kim, H., et al. "Do's and Don'ts: Learning Desirable Skills with Instruction Videos." NeurIPS 2024

---

> ### Comment · Reviewer_AeWi · 2026-01-14
>
> Thank you for your prompt response.
>
> CSD applies an arbitrary non-negative distance function to **all state pairs** in the state space, i.e., for all $(x, y) \in X$. This setting is valid: the function itself does not need to be a proper metric because, as shown in CSD, it implicitly induces a valid pseudo-metric.
>
> The key difference is that this work applies an arbitrary distance function **only to adjacent state pairs**, which constitutes a fundamentally different setting.
>
> METRA applies the constraint only to adjacent pairs because it specifically uses *temporal distance*, which is a constant value of 1. Theorem B.3 shows that, for temporal distance, restricting the constraint to adjacent pairs is theoretically justified.
>
> If this work intends to apply the constraint only to adjacent state pairs, I believe the work must provide a valid justification that such a formulation can indeed induce a valid metric (or at least a pseudo-metric) over all state pairs. Without it, the learned latent space can't be a metric space, which leads to make the whole "distance" maximization training lose its points. I am afraid that this is not only unproven, but also unlikely to hold in general.

---

> > ### Author Response · Authors · 2026-01-15
> > **Response to the concern on the state space**
> >
> > Thank you for this important theoretical discussion. We would like to clarify the theoretical grounding for our approach. Should any aspects remain unclear or if the reviewer has further suggestions on how to better present this theoretical justification, we would be very happy to discuss and incorporate the feedback.
> >
> > 1. CSD [1]: While Theorem 4.1 states the constraint for all state pairs $\forall x, y \in \mathcal{S}$, their actual implementation (Section 4.3) samples $(x,y)$ from the policy's transition distribution $p(s,s')$, i.e., adjacent state pairs only. This is the same practical setup we use.
> >
> > 2. DoDont [2]: Explicitly formulates the constraint on adjacent pairs $\mathcal{S}adj$ (Equation 5), using a binary classifier output $\hat{p}_\psi(s,s') \in [0,1]$ as the distance function. They cite CSD for theoretical validity, noting that "a generic distance function does not necessarily need to meet the criteria of a valid distance metric... the original constraint can be implicitly converted into one with a valid pseudometric."
> >
> > Our score function $f(s') \in [\alpha, 1]$ is non-negative and bounded, satisfying the same conditions as DoDont's instruction network. We apply the constraint on adjacent state pairs, which is consistent with both CSD's implementation and DoDont's formulation.
> >
> > We also note that CSD, DoDont, and our FoG all adopt this same practical setup and have demonstrated reasonable empirical performance in their respective experiments.
> >
> > We sincerely thank the reviewer for their time and thoughtful feedback, and we welcome further discussion.
> >
> > ---
> > [1] Park, S., Lee, K., Lee, Y., & Abbeel, P. "Controllability-Aware Unsupervised Skill Discovery." ICML 2023.
> >
> > [2] Kim, H., Lee, B., Lee, H., Hwang, D., Kim, D., & Choo, J. "Do's and Don'ts: Learning Desirable Skills with Instruction Videos." NeurIPS 2024.

---

> > > ### Comment · Reviewer_AeWi · 2026-01-20
> > >
> > > 1. I first acknowledge that a similar issue exists in CSD [Park et al., 2023]. CSD indeed updates the network using only adjacent state pairs. However, this work predates METRA [Park et al., 2024], and the follow-up work METRA explicitly addresses this limitation by introducing a proper distance metric. LGSD [Rho et al., 2025] likewise relies on a well-defined distance metric. Since the current work can be viewed as a successor to these prior approaches, it is reasonable to expect a similarly principled treatment.
> > >
> > >     The authors also note that DoDon’t updates using only adjacent state pairs. If I were reviewing DoDon’t, I would raise the same concern there as well.
> > >
> > >     My concern is that without a valid distance metric, the optimization procedure may suffer from poor convergence properties. The ultimate goal is to optimize WDM; however, if WDM is defined without a proper distance metric, it becomes unclear what quantity is being optimized. Enforcing constraints only on adjacent pairs cannot ensure globally well defined distance metric. In this sense, although the paper presents successful training results on the selected tasks, it is difficult to assess how robust the method would be when applied to other problem settings not covered in the paper.
> > >
> > >
> > >    That said, let me momentarily set this issue aside and assume that updating only with adjacent pairs does not empirically cause problems across the diverse tasks presented in the paper.
> > >
> > > 2. Under this assumption, my next criterion is the **novelty of the proposed idea**. On this front, I do not find the contribution to be sufficiently novel. The proposed method shares substantial similarities with existing approaches, such as LGSD and DoDon’t. In particular, LGSD already leverages guidance from a large language model, which itself qualifies as a foundation model.
> > >
> > >    According to the authors, the key distinction of this work is the demonstration of the approach on pixel-based tasks. This is indeed a valid contribution, as LGSD is difficult to be applied in pixel-based settings. However, this claim would be significantly strengthened if the authors included LGSD as a baseline and empirically demonstrated that LGSD performs poorly in pixel-based environments while the proposed method succeeds. Since the environments considered in this work still has internal numeric state representations, such as humanoid joint angles, LGSD could still be applied. This makes the absence of a direct comparison more concerning.
> > >
> > > For these reasons, I am leaning toward rejection. That said, if other reviewers feel strongly that this paper should be accepted, I would not oppose the final decision.

---

> > > > ### Author Response · Authors · 2026-01-21
> > > > **Response to Reviewer AeWi**
> > > >
> > > > We thank the reviewer for their continued engagement and thoughtful feedback.
> > > >
> > > > Regarding **novelty**, we note that CSD, DoDont, LGSD, and our FoG all operate within the same distance-maximizing skill discovery framework, differing primarily in their choice of distance metric (controllability-aware distance for CSD, instruction network for DoDont, language-based semantic distance for LGSD, and foundation model-guided score functions for FoG). We expect that more distance metrics will be proposed and inserted into this framework to learn other interesting behaviors. In this sense, our work follows a similar methodology as these prior works, each offering a different perspective on defining "desirable" transitions, and we provide strong empirical results to support our approach.
> > > >
> > > > Regarding the comparison with LGSD, we agree that adding LGSD as a baseline would be interesting. However, we note that our paper already includes 8 baselines, and a direct comparison faces methodological challenges: LGSD operates on ground-truth state representations, while FoG is designed for pixel-based inputs. Running LGSD with state inputs while our method uses pixels would be an unfair comparison.
> > > >
> > > > We will add a discussion further clarifying the relationship with LGSD in the revised manuscript, and we remain open to further suggestions.

---

### Review · Reviewer_B3Th · 2025-11-13

**Summary Of Contributions:**

The authors propose a novel framework for skill discovery that leverages foundation models. In doing so, the proposed framework makes it possible to define rewards for complex skills, which is something that could otherwise be intractable to do without the help of foundation models. Moreover, the authors show that this framework can be used to guide the skill discovery such that the learned skills are aligned with human preferences. Finally, the authors show that through this framework, the aligned skill discovery can be used to avoid undesired behaviours (where ‘undesired’ is with reference to pre-specified human preferences).

**Audience:**

Yes

**Audience Explanation:**

Guiding skill discovery to be aligned with human preferences would be of interest to several members of TMLR’s audience, including those with interests in skill discovery, and more broadly, alignment.

**Broader Impact Concerns:**

It could be beneficial to include a statement that mentions that the proposed method guides skill discovery with respect to human preferences, and that those human preferences are inherently subjective, and could therefore conceivably be in line with unethical objectives/motivations.

**Claims And Evidence:**

Yes

**Claims Explanation:**

The authors make three primary claims in this paper:

1) The proposed method incorporates human intentions into skill discovery through foundation models.
2) The proposed method successfully learns to eliminate undesirable behaviours.
3) The proposed method can discover skills involving behaviours that are “difficult to define”.

In terms of the first claim, this claim is supported by accurate, convincing, and clear evidence. In particular, the authors begin by showing through Figure 1, at a high and conceptual level, how the foundation models are leveraged in their method. Then, in Section 3.1 (with mode details in Appendix A), they explicitly show how the output of the foundation model is incorporated into the underlying skill discovery framework (Equation 7). Finally, the authors provide several examples where they show how foundation models are used in their framework.

In terms of the second claim, this claim is supported by accurate, convincing, and clear evidence. First, Appendix F shows intuitive and clear examples of how the foundation models are able to accurately capture the pre-specified human preferences (for example, in the Ant environment, avoiding to go south). Then the empirical results show that the desired behaviour is ultimately achieved for the shown examples (for example, in the Ant example, Figure 3 shows how none of the learned skills allow the Ant to go south).

In terms of the third claim, this claim is supported by somewhat accurate and clear evidence, but it is not entirely convincing. The example used by the authors to prove their claim invokes the Humanoid environment, where they test whether their method can successfully learn skills that result in the humanoid being “stretched” or “twisted”. Their evaluation method consisted of using 10 human participants to judge whether the humanoid indeed looked “stretched” or “twisted”. In this regard, the evaluation method is inherently subjective and a larger sample size could be more convincing. However, I recognize that the mere premise of this claim (“hard to define”) makes it challenging to have a truly objective evaluation criteria. That being said, it is easy to see that a more rigorous subjective evaluation could be performed.

**Requested Changes:**

From a writing perspective, the main body in the paper is in good shape, with the appendix having a higher frequency of typos. Typos in the main body include:
- The paragraph between Equations 6 and 7 is awkwardly phrased and should be improved
- Section 3.2, State-based section at the very end: Section F.7.1 -> Appendix F.7.1
- Section 4.2 second bullet: the sentence is not complete

In terms of the figures, many of them are hard to read without a y-axis title. I strongly recommend that the authors add a y-axis title to all plots.

Non-formatting/grammar requests are as follows:

Section 2 is missing a connection to foundation models. In particular, where do foundation models fit in within the DSD framework? The authors should make this connection explicitly, as well as formally define foundation models.

The authors claim that one of the benefits of their method is that it reduces human effort. Yet, after reading the paper I am still left wondering what the level of human effort is for their method. I understand that the authors provide examples, but they primarily treat the prompt to the foundation model as a given input. How complex is this prompt? One potential suggestion for the authors is to compare a typical prompt for their method to a manually designed reward function to contrast the level of effort.

Perhaps my biggest concern that should be addressed is that, aside from a very brief paragraph on "score function noise", there is no discussion on the non-determinism of the score functions. In particular, if the authors were to repeat their experiments numerous times, I imagine that the generated score function would fluctuate to some extent given the non-determinism of foundation models. Accordingly, it is conceivable that some generated functions may not work (either there is a bug in the generated code, or it fails to properly encode the preferences, or it hallucinates, etc.). What is the success rate in this regard? How much do the score functions vary given the same input prompt. How sensitive are the score functions with regard to the input prompt (that is, if the input prompt changes a bit, does it drastically change the generated score function)?

---

> ### Author Response · Authors · 2025-12-17
> **Author Response (17/12/25)**
>
> Thank you for your review! We address all your requested changes below:
>
> > The paragraph between Equations 6 and 7 is awkwardly phrased and should be improved.
>
> We now improved the Eq.6 and Eq.7, and polished the sentences between them. Please see the updated manuscript.
>
> > incomplete sentences in Section 3.2 and 4.2.
>
> Thanks for pointing these out, we fixed all of them.
>
> > In terms of the figures, many of them are hard to read without a y-axis title. I strongly recommend that the authors add a y-axis title to all plots.
>
> As you suggested, we now have added the y-axis title to all figures, please see updated manuscript.
>
> > Section 2 is missing a connection to foundation models. In particular, where do foundation models fit in within the DSD framework? The authors should make this connection explicitly, as well as formally define foundation models.
>
> We now discussed related work on "foundation models in RL" in Appendix B and explained the usage of foundation models in Section 3.2.
>
> > How complex is this prompt?
>
> The prompts we used for both state-based and pixel-based tasks are shown in the Appendix F.7. For our main experiments (pixel-based), prompt is just two sentences, describing what we want the agent (not) to do.
>
> > One potential suggestion for the authors is to compare a typical prompt for their method to a manually designed reward function to contrast the level of effort.
>
> We did include a baseline METRA+ that uses a human designed reward function, which shows inconsistent performance, i.e. poor performance in Fig.4 and competitive performance in Fig.5, indicating designing reward functions manually can be difficult sometimes.
>
> > Perhaps my biggest concern that should be addressed is that, aside from a very brief paragraph on "score function noise", there is no discussion on the non-determinism of the score functions. In particular, if the authors were to repeat their experiments numerous times, I imagine that the generated score function would fluctuate to some extent given the non-determinism of foundation models. Accordingly, it is conceivable that some generated functions may not work (either there is a bug in the generated code, or it fails to properly encode the preferences, or it hallucinates, etc.). What is the success rate in this regard? How much do the score functions vary given the same input prompt. How sensitive are the score functions with regard to the input prompt (that is, if the input prompt changes a bit, does it drastically change the generated score function)?
>
> For state-based tasks, we used ChatGPT or Claude, and since the given task is simple and the environment context is given, we only prompted it once and the output function at the first time was correct. We believe that the modern commercial LLMs (like ChatGPT, Claude, Gemini, etc) are good enough to output a correct simple score function in the first try.
>
> To evaluate the prompt robustness of FoG for pixel-based tasks, which is essentially the prompt robustness of the foundation model (CLIP) we are using for the pixel-based tasks, we have newly conducted the same experiments with different prompts.
>
> In Cheetah flip, we tested: "flip over", "roll over", "upside down", "upend" as the undesirable textual descriptions. See results in the table below. Although these prompts mean similarly to us, they have quite different performance. E.g. “flip over” and  “roll over” have the best and similar performance, while “upend” and “turn upside down“ perform poorly. So we do suggest play around with the prompt when applying FoG to your own tasks.
>
> | | Flip over | Roll over | Upend | Turn upside down |
> | :--- | :--- | :--- | :--- | :--- |
> | % Flip &darr; | $0.11 \pm 0.05$ | $0.04 \pm 0.02$ | $0$ | $0.74 \pm 0.05$ |
> | State coverage &uarr; | $29$ | $21 \pm 0.47$ | $3$ | $29 \pm 0.47$ |
>
> We also added a figure to the ablation study to show these results (please see updated Section 4.3 and newly added Fig.8 in the manuscript).

---

### Review · Reviewer_ELTc · 2025-12-12

**Summary Of Contributions:**

The paper proposes using foundation models to weight reward functions. They show that this allows learning of diverse skills that are better aligned with human intentions and can satisfy constraints better than existing skill discovery methods. Foundation models are used to generate (a) the reward function when the states are known or (b) are used to map pixels to embeddings and a simple cosine distance between the desired and undesired text is used as the reward weight. Several experiments are conducted with different agents of varying complexity for both state-based and pixel-based cases. The results show that the proposed method generally matches or outperforms several baselines, including those that need human demonstrations.

**Audience:**

Yes

**Audience Explanation:**

Yes, skill discovery for agents is a fundamental topic in artificial intelligence. This paper uses foundation models to improve skill discovery, and is likely to be interesting to a significant part of TMLR's audience.

**Broader Impact Concerns:**

No concerns.

**Claims And Evidence:**

No

**Claims Explanation:**

I am not very familiar with this sub-field of research, but the experimental settings seems a little too simple. They are all fully synthetic and are not very realistic environments.

I am especially underwhelmed by the initial experiments on state-based tasks. The score functions are so simple that they could be easily written by a human. I don't understand why foundation models would be needed here. The authors should try to demonstrate this on a more difficult task.

Equation (7) and the derived Equation (11) in the appendix don't match. Equation (7) shows $\phi(s)$ in the objective function, but it should be $\tilde{\phi}(s)$ according to Equation (9). Please check the notation, and confirm that the experimental results are using the accurate formulation.

**Requested Changes:**

I think the writing of the paper needs improvement. Particularly, as I don't have direct experience with this field of research, I would appreciate a few extra details about the methods.

1. There should be additional details on the parametric form of the mapping from the latent vector $z$ to the policy. I am a little confused about it.
2. What exactly is the parametric form of the mapping $\phi$?
3. The objective function is Equation (2) is not clear to me. What is the motivation for this? Why is it important to align $z$ with the difference of the mapped consecutive states? Why does it lead to learning diverse skills? This is crucial for understanding the method.
4. Equation (7) is not correct according to the derivation in the appendix, as mentioned earlier. Please check.

---

> ### Author Response · Authors · 2025-12-17
> **Author Response Part 1 of 2 (17/12/25)**
>
> Thank you for your review! We first address all your requested changes and then go through all questions you had below:
>
> **Requested Changes**
>
> > There should be additional details on the parametric form of the mapping from the latent vector z to the policy. I am a little confused about it.
>
> A latent vector z is first sampled, then the RL agent is taking it as a condition to form a ‘skill-conditioned’ policy, $\pi(a|s,z)$, which is a standard MLP. The policy network $\pi(a|s,z)$ takes the concatenation of the state observation $s$ and the skill vector $z$ as its input. We have also added a few sentences of explanations at the beginning of Section 2, and please see the updated manuscript.
>
> > What exactly is the parametric form of the mapping $\phi(s)$?
>
> $\phi(s)$ is a representation function, which is a neural network encoder.
>
> > The objective function is Equation (2) is not clear to me. What is the motivation for this? Why is it important to align  with the difference of the mapped consecutive states? Why does it lead to learning diverse skills?
>
> Unlike some prior methods that maximize mutual information objective (DIAYN[1]), WDM objective (which LSD[2], CSD[3], METRA[4], and our method FoG use) stems from the dual formulation of the Wasserstein distance.
>
> The original objective is defined on the initial and final state distribution, which can be decomposed into the telescoping sum form, i.e. Eq.2, aligning $z$ with the difference of the mapped conservative states turns the global objective into a dense objective using a telescoping sum, which makes sure that the agent is rewarded step-by-step.
>
> The metric $d$ in Eq.2 enforces an upper bound on latent transitions so that differences in the latent space do not exceed the distance measured by $d$. Under this constraint, the RL agent learns to maximize $\phi(s_{t+1}) - \phi(s_{t})$ in certain directions $z$, thereby discovering diverse skills that traverse the largest distances measured by the distance metric $d$ in the latent space.
>
> > Equation (7) is not correct according to the derivation in the appendix, as mentioned earlier. Please check.
>
> Thanks for pointing it out. We now updated the Eq. 7 and Eq. 8, and improved the derivation in the Appendix A.
>
> -----
>
> [1] Eysenbach, Benjamin, et al. "Diversity is all you need: Learning skills without a reward function."
>
> [2] Park, Seohong, et al. "Lipschitz-constrained unsupervised skill discovery."
>
> [3] Park, Seohong, et al. "Controllability-aware unsupervised skill discovery."
>
> [4] Park, Seohong, Oleh Rybkin, and Sergey Levine. "Metra: Scalable unsupervised rl with metric-aware abstraction."

---

> ### Author Response · Authors · 2025-12-17
> **Author Response Part 2 of 2 (17/12/25)**
>
> **Questions and Concerns**
>
> > I am not very familiar with this sub-field of research, but the experimental settings seems a little too simple. They are all fully synthetic and are not very realistic environments.
>
> All tasks used in our work are taken from commonly-used beachmarks in the unsupervised skill discovery research field, which are also consistent with other work in the field like DIYAN[1], LSD[2], METRA[3], DoDont[4].
>
> > I am especially underwhelmed by the initial experiments on state-based tasks. The score functions are so simple that they could be easily written by a human. I don't understand why foundation models would be needed here. The authors should try to demonstrate this on a more difficult task.
>
> The experiments on state-based tasks act as a proof-of-concept, i.e. to prove that the proposed score function pipeline does work for state-based tasks. And our main experiments focus on the pixel-based tasks, which are also more challenging.
>
> > Equation (7) and the derived Equation (11) in the appendix don't match. Equation (7) shows  in the objective function, but it should be  according to Equation (9). Please check the notation, and confirm that the experimental results are using the accurate formulation.
>
> Since we optimize exclusively over this new scaled representation $\tilde\phi$, for notational simplicity, we rename $\tilde\phi$ to $\phi$ in the final objective. We also added an extra explanation in Appendix A.
>
> --------
>
> 1] Eysenbach, Benjamin, et al. "Diversity is all you need: Learning skills without a reward function."
>
> [2] Park, Seohong, et al. "Lipschitz-constrained unsupervised skill discovery."
>
> [3] Park, Seohong, Oleh Rybkin, and Sergey Levine. "Metra: Scalable unsupervised rl with metric-aware abstraction."
>
> [4] Kim, Hyunseung, et al. "Do's and Don'ts: Learning Desirable Skills with Instruction Videos."

---

### Review · Reviewer_tSPs · 2025-12-19

**Summary Of Contributions:**

- Proposes FoG, a method that extracts score functions from foundation models (LLMs for state-based tasks, CLIP for pixel-based) to re-weight METRA's skill discovery rewards, enabling learning of diverse skills aligned with human preferences
- Make an attempt at showing theoretical equivalence between reward re-weighting with the score function and using it as the distance metric in the Distance-maximizing Skill Discovery (DSD) objective
- Evaluates against 7 baselines (METRA, METRA+, LSD, DoDont, DoDont+, FR-SAC, METRA+FR) across 5 environments: 2 state-based (HalfCheetah, Ant) and 3 pixel-based (Cheetah, Quadruped, Humanoid)
- Shows FoG reduces undesirable behaviors (e.g., flipping reduced from >50% to ~0% in Cheetah) and enables avoidance of designated hazardous areas while maintaining state coverage
- Claims FoG can discover skills involving subjectively-defined behaviors (e.g., "twisted" postures in Humanoid), validated via human study (90% preference for FoG)

**Audience:**

Yes

**Audience Explanation:**

LLMs have significant potential for option and skill discovery, particularly in the context of alignment, and this paper takes a step in that direction. As a result, the work is likely to be of interest to part of the TMLR and broader RL community. That said, the paper would benefit from a more cohesive narrative, as the state-based and pixel-based approaches rely on quite different mechanisms and are not clearly unified or motivated as a single framework.

**Broader Impact Concerns:**

No broader impact concerns.

**Claims And Evidence:**

No

**Claims Explanation:**

I thank the authors for their contributions and for engaging with the review process. I have reviewed their responses to other reviewers, and while some concerns were addressed, several important issues remain.
1. In Section 5, the paper argues that LGSD is computationally expensive due to step-wise LLM querying and implicitly claims FoG is more efficient by extracting a score function applied once in state-based settings or via batch processing in pixel-based settings. However, LGSD is not included as a baseline, so this claim is not empirically validated. This is particularly important for pixel-based environments: although FoG avoids online LLM calls, it still requires repeated CLIP inference over observations, even if batched, making the actual compute or speed advantage over LGSD unclear without direct comparison. Given that both methods rely on foundation models and target alignment through reward shaping, directly comparing their performance and efficiency gaps is important, rather than assuming FoG’s superiority.
2. The experiments are reported using only three random seeds, with error bars that appear to be standard error, which is not sufficient for statistical significance. Prior work has shown that deep RL results can vary substantially across seeds and that small-seed evaluations are often unreliable [1,2]. This is especially concerning given that METRA, one of the main baselines, reports results over eight seeds, indicating a mismatch in evaluation rigor. Increasing the number of seeds and clearly defining the reported statistics would strengthen the empirical claims.
3. The score functions shown in the paper, particularly in the state-based setting, are very simple and could be easily written by a human without the use of a foundation model. In their response, the authors clarify that these state-based experiments are intended only as a proof-of-concept, with the main contribution lying in the pixel-based setting. While this clarification is reasonable, it is not clearly reflected in the paper’s narrative. As written, the paper presents state-based and pixel-based results as parallel instantiations of the same framework, which implicitly suggests that foundation models play a meaningful role in both settings. If the state-based experiments are merely a sanity check, the story of the paper should be adjusted accordingly to avoid overstating the role of foundation models in simple settings.
4. A related but secondary issue is that the mechanisms used for state-based and pixel-based environments are quite different in practice. The state-based setting relies on LLM-generated, code-level score functions, while the pixel-based setting uses CLIP-style embedding similarity that functions more like a classifier. Although both are framed under the same “score function” abstraction and used for reward shaping, the underlying mechanisms and assumptions are substantially different, making the claimed unification between the two settings somewhat unclear.

------
**References**:

[1] Henderson et al., Deep Reinforcement Learning That Matters, AAAI 2018.

[2] Agarwal et al., Deep Reinforcement Learning at the Edge of the Statistical Precipice, NeurIPS 2021.

**Requested Changes:**

### Major changes:
1. To address statistical significance with a low number of random seeds, the authors should consider applying the methodology proposed in Deep Reinforcement Learning at the Edge of the Statistical Precipice [1], which provides principled techniques (e.g., paired comparisons and bootstrap-based analyses) for obtaining statistically meaningful conclusions even with few seeds. That said, the experimental evaluation would be substantially stronger if the authors increased the number of independent runs to at least 8 (or more), consistent with prior work such as METRA, rather than relying solely on statistical corrections for small-sample regimes.
2. To improve empirical coverage and ensure fairer comparisons, it would be beneficial for the authors to include at least one additional environment, such as Kitchen, which was used and evaluated in the METRA paper [2] and is already referenced in the comparison narrative. Including this environment would better contextualize the performance claims. That said, I would be willing to overlook this omission if the more critical issues are properly addressed.
3. The paper should include stronger and more relevant baselines, particularly LGSD [3]. LGSD is discussed in the related work and appears to be one of the most conceptually similar approaches to the proposed method, yet it is not included in the experimental comparisons. This omission weakens the empirical positioning of the contribution.
4. I would appreciate it if the authors could clarify the narrative in the paper around state-based versus pixel-based settings, in particular by explaining why the state-based score functions and rewards are intentionally simple, how this design choice aligns with the paper’s goals and scope, and how this simplicity should be interpreted when comparing against prior work.
5. I would also appreciate a brief but explicit justification for relying exclusively on **CLIP** in the pixel-based setting (and in the ablations), rather than other foundation or vision-language models, for example, in terms of reproducibility, computational cost, stability, or ease of integration, and clarification on how this choice affects the generality of the conclusions.


### Minor changes:
* Clarify what the error bars represent (SD vs SEM vs CI) and how they are computed, especially given only 3 random seeds.
* Fix the baseline count inconsistency: the text claims six baselines but lists seven (in the new revision).
* Fix inconsistent naming of LGSD vs LSGD, including in Table 1.
* Standardize terminology (e.g., “cosine similarity” vs “Cos similarity”) across the paper and appendix.

**References**:

[1] Agarwal et al., *Deep Reinforcement Learning at the Edge of the Statistical Precipice*, NeurIPS 2021.

[2] Park et al., *METRA: Scalable Unsupervised Reinforcement Learning with Metric-Aware Abstraction*, ICLR 2024.

[3] Rho et al., *Language-Guided Skill Discovery*, ICLR 2025.

---

> ### Author Response · Authors · 2025-12-23
> **Author Response Part 1 of 2 (23/12/25)**
>
> Thank you for your review! We address all your requested changes below:
>
> **Major Changes**
> > To address statistical significance with a low number of random seeds, the authors should consider applying the methodology proposed in Deep Reinforcement Learning at the Edge of the Statistical Precipice [1], which provides principled techniques (e.g., paired comparisons and bootstrap-based analyses) for obtaining statistically meaningful conclusions even with few seeds. That said, the experimental evaluation would be substantially stronger if the authors increased the number of independent runs to at least 8 (or more), consistent with prior work such as METRA, rather than relying solely on statistical corrections for small-sample regimes.
>
> We have updated our main results for FoG with 5 additional seeds, now in total 8 independent runs, as shown in updated Fig. 4 and 5 in the revised manuscript. The performance of FoG remains stable, results with 8 seeds closely match those with 3 seeds. Experiments with additional seeds for the remaining baselines are currently running and will be updated upon completion.
>
> > To improve empirical coverage and ensure fairer comparisons, it would be beneficial for the authors to include at least one additional environment, such as Kitchen, which was used and evaluated in the METRA paper [2] and is already referenced in the comparison narrative. Including this environment would better contextualize the performance claims. That said, I would be willing to overlook this omission if the more critical issues are properly addressed.
>
> We previously included preliminary experiments on Kitchen in Appendix E.2 (e.g., biasing the arm towards left/right/stretched), but found evaluation challenging, so we left it for future work.
>
> We have now added another preliminary experiment on Kitchen in Appendix E.2: since we found that METRA learns to open the cabinet ~70% of the time, we prompted FoG to "not open the cabinet." FoG successfully reduces cabinet opening to 22%, and we see clearly FoG learns to suppress this behavior. Videos comparing METRA and FoG are available on our [website](https://sites.google.com/view/submission-fog/preliminary-results-on-kitchen)
>
> We acknowledge that FoG's current performance is limited on complex tasks like Kitchen that require spatial understanding, as CLIP was not trained for such reasoning. We leave more sophisticated vision-language backbones for future work.
>
> > The paper should include stronger and more relevant baselines, particularly LGSD [3]. LGSD is discussed in the related work and appears to be one of the most conceptually similar approaches to the proposed method, yet it is not included in the experimental comparisons. This omission weakens the empirical positioning of the contribution.
>
> LGSD requires a multi-stage pipeline to compute semantic distances between states: (1) a rule-based annotator converts low-dimensional state vectors into templated text (e.g., [0.3, 0.2, 0.11, 0.3] → "object at [0.3, 0.2], robot at [0.11, 0.3]"), (2) GPT-4 is queried to generate semantic descriptions for each discretized state, (3) descriptions are embedded via Sentence-Transformer, and (4) cosine distance between embeddings defines the language distance.
>
> We did not include LGSD as a baseline due to fundamental differences in problem setting and practical constraints:
>
> 1. Input modality mismatch: LGSD operates exclusively on low-dimensional state vectors (e.g., robot proprioception, object positions). It relies on a rule-based state-to-text annotator that converts structured state vectors into text templates (e.g., [0.3, 0.2] → "object at position [0.3, 0.2]"). Our method operates on image observations, where such rule-based annotators are not applicable, one cannot write hand-crafted rules to convert raw pixels into meaningful text descriptions.
> 2. Computational cost: LGSD queries GPT-4-Turbo for every unique (discretized) state to generate semantic descriptions. Even with caching, this incurs substantial API costs that scale with environment complexity, making large-scale experiments expensive to reproduce.
> 3. The LGSD codebase is not publicly available yet.
>
> For clarification, we have also added an improved explanation of LGSD in our related work section, please see the updated Section 5 in the revised manuscript.

---

> > ### Author Response · Authors · 2025-12-23
> > **Author Response Part 2 of 2 (23/12/25)**
> >
> > > I would appreciate it if the authors could clarify the narrative in the paper around state-based versus pixel-based settings, in particular by explaining why the state-based score functions and rewards are intentionally simple, how this design choice aligns with the paper’s goals and scope, and how this simplicity should be interpreted when comparing against prior work.
> >
> > The state-based experiments serve as a proof-of-concept, demonstrating that our pipeline (using a "desirability" score function to re-weight skill discovery rewards) generalizes to state-based tasks. The simplicity of the generated score functions is intentional: since FoG re-weights rather than replaces skill discovery rewards, a binary signal suffices. Importantly, even these simple outputs require the foundation model to automatically identify relevant state dimensions from high-dimensional spaces (e.g., selecting the front tip angle from 18-D observations).
> >
> > Our primary focus is the pixel-based setting, where only image observations are available for learning. We welcome further discussion if any points remain unclear.
> >
> > > I would also appreciate a brief but explicit justification for relying exclusively on CLIP in the pixel-based setting (and in the ablations), rather than other foundation or vision-language models, for example, in terms of reproducibility, computational cost, stability, or ease of integration, and clarification on how this choice affects the generality of the conclusions.
> >
> > We use CLIP due to its widespread adoption, it is open-source with readily available pre-trained weights and lightweight to run (the variant we use,  [clip-vit-large-patch14](https://huggingface.co/openai/clip-vit-large-patch14), has only 0.4B parameters). Importantly, our framework is not tied to CLIP specifically: FoG is designed to work with any vision-language model that provides image-text embeddings which can be used to compute similarity scores (Equation 9). We expect that stronger or task-specific vision-language models would further improve FoG's performance. We leave exploration of alternative backbones for future work. We also have added an explicit discussion of this design choice in the revised paper to clarify the generality of our framework, please see updated Section 6.
> >
> > **Minor Changes**
> >
> > > Clarify what the error bars represent (SD vs SEM vs CI) and how they are computed, especially given only 3 random seeds.
> >
> > The error bars represent standard error of the mean (SEM) computed across independent random seeds. As noted above, we have now updated our main results for FoG to 8 seeds (Figures 4 and 5), which provides more reliable estimates.
> >
> > > inconsistency of naming, terminology.
> >
> > Thanks for pointing these out, we have fixed all of these. Please see the revised manuscript.

---

> > > ### Author Response · Authors · 2026-01-13
> > > **Updated baseline results with additional seeds**
> > >
> > > Following up on our previous reply, we have updated the baseline results in the revised manuscript (Fig. 4 and 5) with 5 additional seeds (now in total 8). The results remain consistent with our initial 3-seed evaluation.

---

### Decision · Action_Editor_FRA3 · 2026-01-28

**Recommendation:** Reject

**Additional Comments:**

This manuscript addresses the challenge of guiding skill discovery toward aligning with human intentions. While prior skill discovery methods focus on maximizing skill diversity, this can result in undesirable behaviors. The authors propose a framework that leverages foundation models (LLMs) to incorporate human intentions, expressed as natural language descriptions, into the skill discovery process. For this, they present a pipeline for integrating current LLMs with existing unsupervised skill discovery methods. Empirical results from multiple low-dimensional ("state-based") and pixel-based RL-type environments demonstrate that this approach reduces undesirable behavior while facilitating the discovery of skills that are "difficult to define".

---

Four reviews have been collected. All reviewers see relevant contributions in this manuscript; key contributions and **strengths** of this work include:

* (S1) a pipeline for integrating current LLMs with existing unsupervised skill discovery methods

* (S2) empirical demonstration of successfully incorporating human intention and thus limiting undesired behavior

* (S3) discovering behaviors that are otherwise difficult to define



At the same time, the reviewers point out substantial shortcomings in the manuscript, affecting clarity and evidence of the results. Based on the evaluation, main **weaknesses** include:

* (W1) More compelling use case needed for the method to be of (stronger) interest to the community. In particular, the motivation of using foundation models for state-based tasks is unclear (as the problem is rather simple then, probably not requiring an LLM) [AeWi, tSPs, B3Th, ELTc]
* (W2) The proposed method shares substantial similarities with existing approaches (e.g., LGSD and DoDon't), which also utilize LLMs. The current presentation does not clearly convey the improvements over these existing methods. [AeWi]
* (W3) Lack of comparison against stronger baselines (e.g., LGSD) [AeWi, tSPs]
* (W4) Concerns about technical results, in particular, the validity of the distance metric [AeWi]
* (W5) Issue of non-determinism of the score functions not properly addressed [B3Th]
* (W6) Presentation needs improvement in different parts, including (but not necessarily limited to) [tSPs, B3Th, ELTc]
  * Section 2 (problem formulation) needs improvement, e.g., missing connection to FM
  * Mechanisms for state-based and pixed-based problems and their difference should be better presented
  * General writing



From my own reading, I would like to add that I found the problem formulation in Section 2 hard to follow and recommend revising it according to the following points (in addition to the reviewers' comments):

* The core components of the method should be introduced as preliminaries and - where applicable - formalized, including: DSD (maximization problem is not stated); what is the definition of phi (a map S-> Z?, "which is the same as the skill space Z" is imprecise); what is the meaning of phi; foundation model
* What precisely is the problem that is solved in the manuscript? This should be explicitly stated in Section 2.
* It might be helpful to disentangle background/preliminaries and problem formulation more. Is the problem that is solved new in itself? Or is it a problem that has been addressed, but is addressed differently herein?



Finally, there was the suggestion that a statement on broader impact should be included [B3Th].

---

Regarding the **review process**, the reviewers provided valuable comments, and authors and reviewers engaged in constructive discussions. While some aspects could be addressed in the discussion, not all weaknesses could be resolved. In the end, most reviewers were leaning reject of the manuscript in the current form.

**Summary of evaluation:** While the manuscript contains results that are potentially of interest and value to the TMLR community, the manuscript in its current form is not ready to be accepted for TMLR based on the above points. If the authors decide to resubmit the manuscript at a later point, they should fully revise the manuscript taking the above points and all reviewers' comments into account.

**Audience:**

Yes

**Audience Explanation:**

LLMs hold significant potential for skill discovery and alignment, which is of interest to part of the TMLR community.

**Claims And Evidence:**

No

**Claims Explanation:**

While the reviewers agree that some of the main contributions are well justified, other claims are not sufficiently evidenced and clear based on the manuscript in its present form. This includes technical aspects (see W4, W5 below), a compelling use case (W1), the need for a clearer distinction from existing methods (W2-W3), and the overall presentation (W6).

**Resubmission Of Major Revision:**

The authors may consider submitting a major revision at a later time.